# Leukocyte immunoglobulin-like receptor B1 (LILRB1) protects human multiple myeloma cells from ferroptosis by maintaining cholesterol homeostasis

Miao Xian [1,6], Qiang Wang [1,6], Liuling Xiao[1], Ling Zhong[1], Wei Xiong [1], Lingqun Ye[1], Pan Su[1], Chuanchao Zhang[1], Yabo Li [1], Robert Z. Orlowski [2], Fenghuang Zhan [3], Siddhartha Ganguly[4], Youli Zu[5], Jianfei Qian[1] & Qing Yi [1] ✉

Multiple myeloma (MM) is a hematologic malignancy characterized by uncontrolled proliferation of plasma cells in the bone marrow. MM patients with aggressive progression have poor survival, emphasizing the urgent need for identifying new therapeutic targets. Here, we show that the leukocyte immunoglobulin-like receptor B1 (LILRB1), a transmembrane receptor conducting negative immune response, is a top-ranked gene associated with poor prognosis in MM patients. LILRB1 deficiency inhibits MM progression in vivo by enhancing the ferroptosis of MM cells. Mechanistic studies reveal that LILRB1 forms a complex with the low-density lipoprotein receptor (LDLR) and LDLR adapter protein 1 (LDLRAP1) to facilitate LDL/cholesterol uptake. Loss of LILRB1 impairs cholesterol uptake but activates the de novo cholesterol synthesis pathway to maintain cellular cholesterol homeostasis, leading to the decrease of anti-ferroptotic metabolite squalene. Our study uncovers the function of LILRB1 in regulating cholesterol metabolism and protecting MM cells from ferroptosis, implicating LILRB1 as a promising therapeutic target for MM patients.

Multiple myeloma (MM) is a hematologic malignancy characterized by the uncontrolled clonal proliferation of plasma cells in the bone marrow (BM)[1]. As the second most common hematological disease after non-Hodgkin lymphoma, MM causes about 100,000 deaths each year worldwide[2]. Despite the demonstrated benefits of novel therapies, relapses are frequent, and acquired resistance to MM treatment eventually emerges in most, if not all, patients. MM patients who suffer more aggressive progression usually result in poorer survival. The

genes driving such unfavored outcomes in MM have not been fully understood. Therefore, there is an urgent need to identify the genes and mechanisms that contribute to the aggressive behaviors of MM in order to develop improved therapeutic strategies for the disease.

To discover potential therapeutic targets for MM patients, we analyzed MM patient data and identified LILRB1, an immunoreceptor tyrosine-based inhibition motif (ITIM)-containing transmembrane protein which belongs to the leukocyte immunoglobulin-like receptor

[1]Center for Translational Research in Hematological Malignancies, Houston Methodist Neal Cancer Center, Houston Methodist Research Institute, Houston, TX 77030, USA. [2]Department of Lymphoma and Myeloma, The University of Texas MD Anderson Cancer Center, Houston, TX 77030, USA. [3]Myeloma Center, Winthrop P. Rockefeller Institute, Department of Internal Medicine, University of Arkansas for Medical Sciences, Little Rock, AR 72205, USA. [4]Houston Methodist Neal Cancer Center, Houston Methodist Research Institute, Houston, TX, USA. [5]Department of Pathology and Genomic Medicine, Institute for Academic Medicine, Houston Methodist Research Institute, Houston, TX 77030, USA. [6]These authors contributed equally: Miao Xian, Qiang Wang. ✉e-mail: qyi@houstonmethodist.org

(LILR) family[3,4], as a promising target for aggressive MM. LILRB1 on immune cells was reported to bind to its ligand, major histocompatibility complex (MHC) class I subunit β2-microgolbulin (B2M), on tumor cells[5]. This interaction leads to immune suppression responses in various immune cells including macrophages[6,7], T cells[8], NK cells[9–11], dendritic cells[12,13], and B cells[14–16]. For instance, disruption of LILRB1 on macrophages promotes the phagocytosis of tumor cells by macrophages[6]. LILRB1 effectively competes with CD8 for MHCI binding, raising the possibility that LILRB1 inhibits CD8+ T cell activation by blocking CD8 binding[8]. Furthermore, interaction of HLA-G on the surface of tumor cells with LILRB1 on NK cells confers protection against NK cell cytolysis[17]. However, the function of tumor-derived LILRB1 in tumor biology remains unclear. Interestingly, abnormally high levels of serum B2M were detected in most MM patients and thus are considered a biomarker for staging and prognosis of MM[18]; nevertheless, the role of LILRB1 in MM has been poorly studied.

Ferroptosis is a type of programmed cell death that depends on iron and is accompanied by a large amount of lipid peroxidation accumulation[19–21]. Analysis of BM aspirates from Monoclonal Gammopathy of Undetermined Significance (MGUS), smoldering MM (SMM), and MM patients identified significant decreases in key polyunsaturated fatty acids (PUFA), including arachidonic acid (AA), which was reported to induce ferroptotic cell death, indicating that ferroptosis is involved in MM progression[22–24]. However, the underlying mechanisms of ferroptosis and MM progression have not been elucidated.

Known metabolic rearrangements in MM cells include adjustments in fatty acid/cholesterol synthesis and degradation[25]. Interestingly, hypocholesterolemia is seen in MM patients[26]. Total cholesterol, low-density lipoprotein (LDL) cholesterol, and high-density lipoprotein (HDL-C) in the blood of MM patients are significantly lower than those of healthy donors, indicating the increased LDL clearance and utilization of cholesterol by MM cells[26–28]. These findings suggested that cholesterol metabolism is important for MM cells. As lipids are the substrates of lipid peroxidation, and their metabolisms regulate the sensitivity of cells to ferroptosis[29–31], ferroptosis is closely related to lipid metabolism. Yet, the crosstalk between metabolic reprogramming and ferroptosis in MM cells has not been clarified clearly, nor how it contributes to MM progression.

In this study, we performed a series of in vitro and in vivo experiments with human MM cell lines and murine MM models to explore the function of LILRB1 in MM progression and revealed the underlying mechanisms of LILRB1 in regulating MM ferroptosis by facilitating LDL/cholesterol uptake. We uncovered a role of LILRB1 in maintaining cholesterol homeostasis and established it as a promising target for MM therapy.

## Results

### LILRB1 is one of the top 20 upregulated genes in MM patients with poor prognosis

To identify genes driving an aggressive MM, we compared gene expression profiles (GEP) of MM cells between MM patients with good prognosis (survival ≥4 years from diagnosis), and MM patients with poor prognosis (survival <2 years from diagnosis), using Zhan et al. datasets[32]. More than 7000 significantly and differentially expressed genes ($P < 0.05$) were identified (Fig. 1a). Among the top 20 upregulated genes in the MM patients with survival <2 years, CCR10[33], PHF19[34], FOXM1[35], DSG2[36], and others, have been reported to be associated with MM aggressive progression and poor prognosis (Fig. 1a), supporting the reliability and rationality of our analysis. The expression of the top 20 upregulated genes among MM patients with poor survival is shown (Fig. 1b). The variability in gene expression among patients demonstrates the complexity and diversity of human biology and disease. To our interest, LILRB1, an immune response-related transmembrane protein, was ranked 16th among the 2714

genes upregulated in MM patients with survival <2 years (Fig. 1b and Supplementary Data 1). LILRB1 is a member of the immune inhibitory receptor family LILRBs, which are expressed on cell surface of immune cells and can contribute to immune evasion[4]. Recently, other members of the LILRB family, such as LILRB3[37,38] and LILRB4[39], have been reported to support the survival of cancer cells. Interestingly, B2M, one of the ligands of LILRB1, was found to be highly elevated in the serum of most MM patients[40–42], suggesting that LILRB1 may play an important role in MM progression. Consequently, we decided to further investigate the potential of LILRB1 as a target and player in human MM.

### High expression of LILRB1 correlates with poor prognosis in MM patients

To determine whether LILRB1 is related to aggressive behaviors of human MM, we analyzed several different MM patient datasets from Oncomine[43]. The Carrasco myeloma dataset[44] showed that patients with recurrent MM had a higher expression of LILRB1 than non-recurrent MM patients (Fig. 1c). In Zhan et al.'s dataset[45], MM cells from MGUS and MM patients expressed significantly higher levels of LILRB1 than normal plasma cells from healthy donors (Fig. 1d). Moreover, Broyl's MM dataset[46] showed that MM patients in advanced stage III had a significantly higher LILRB1 expression compared to those in early stages (I and II) (Fig. 1e). Interestingly, MMRF CoMMpass study IA13 datasets showed that MM cells from patients with high-risk cytogenetic abnormality t(4;14) translocations, which are related to poor prognosis[47,48], had a higher LILRB1 expression than those from patients with standard-risk cytogenetics (Fig. 1f). Furthermore, MM patients with a higher expression of LILRB1 had inferior survival rates than patients with a lower expression of LILRB1 (Fig. 1g, h). Together, these data revealed that high LILRB1 expression is highly associated with the aggressive behaviors of MM and contributes to MM progression and poor survival.

Next, we determined LILRB1 expression in human MM cells. Compared with normal human B cells, MM cell lines ARP-1, MOLP-8, NCI-H929, and MWD MM.13 showed high LILRB1 expression, while other cell lines had low protein expressions (Fig. 1i). The expression of LILRB1 demonstrated a heterogeneity across patient samples (Fig. 1j and Supplementary Fig. 1a), with a common occurrence in primary patient myeloma cells. While most (21/24) primary MM patient samples exhibited the expression of LILRB1, the expression levels of LILRB1 varied within each sample. As LILRB1 is commonly and highly expressed by human MM cells from patients, we considered it as a potential target that warrants further investigation.

### Knockdown (KD) of LILRB1 slows MM development and reduces tumor burden in vivo

To determine the role of LILRB1 in human MM, we knocked down LILRB1 in three human MM cell lines that express high levels of LILRB1 (Supplementary Fig. 1b). Compared to control (CTR)-KD cells, LILRB1-KD MM cells had similar apoptotic rates in vitro (Supplementary Fig. 1c). We injected luciferase-expressing CTR-KD or LILRB1-KD MM cells into NSG mice via the tail vein to examine the function of LILRB1 in MM cells in vivo. NSG mice bearing LILRB1-KD MM cells displayed significantly slower tumor growth (Fig. 2a–d), lower tumor burden (Fig. 2e, f), impaired tumor infiltration into BM (Fig. 2g, h), and better survival (Fig. 2i, j) compared to CTR-KD MM-bearing mice. We also overexpressed LILRB1 in MM.1R MM cells (Supplementary Fig. 2a), which express a low level of LILRB1 (Fig. 1i), and confirmed the effect of LILRB1 overexpression in MM cells in vivo (Supplementary Fig. 2b, c). LILRB1-overexpressing MM.1R-bearing mice showed significantly higher tumor burden (Fig. 2k), more tumor infiltration into BM (Fig. 2l), and poorer survival (Fig. 2m) compared to CTR MM.1R-bearing mice. These data revealed that LILRB1 may play an important role in MM development and progression, and indicated that LILRB1 may be a potential therapeutic target for human MM.

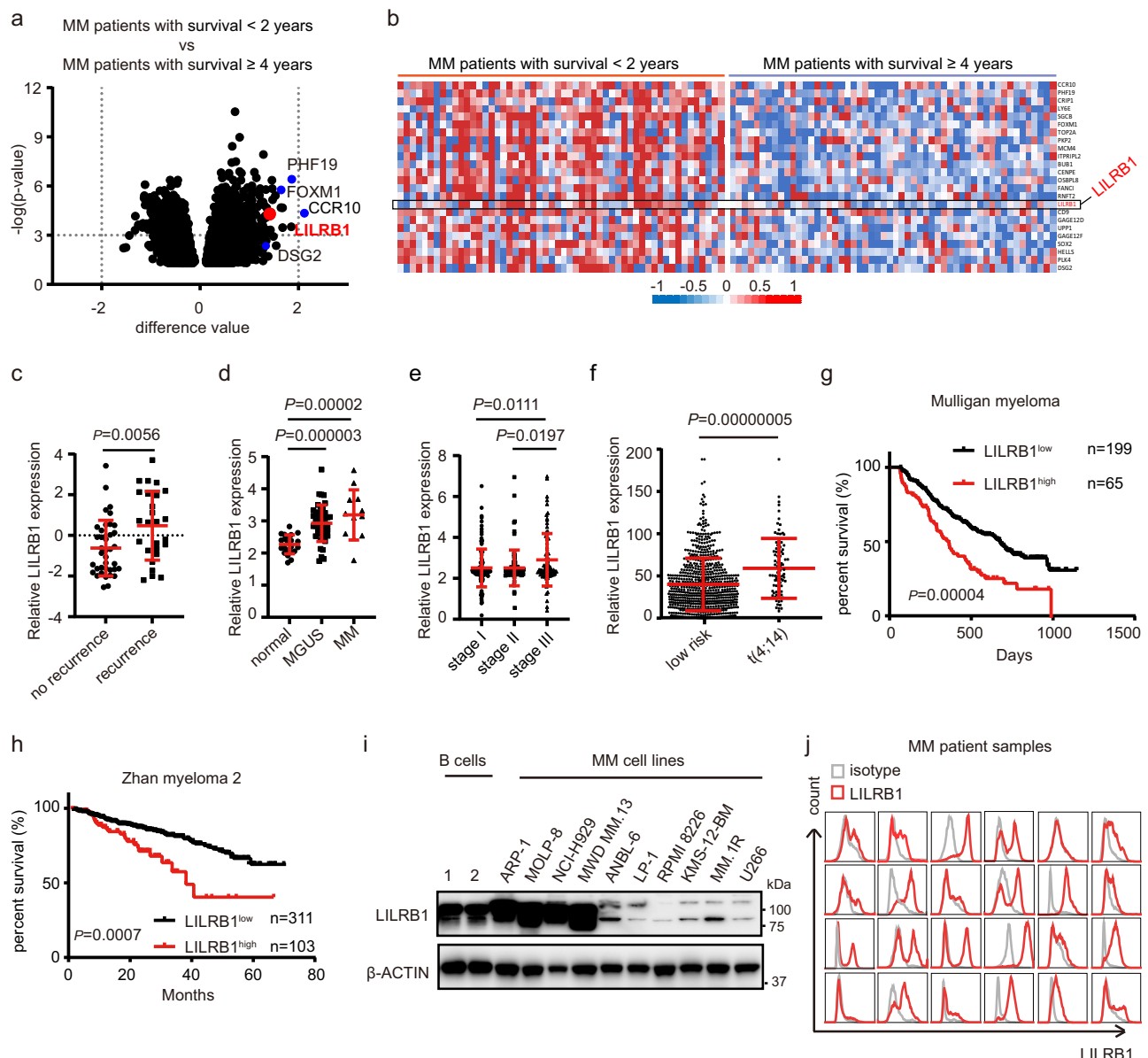

**Fig. 1 | High expression of LILRB1 is closely related to aggressive behaviors of MM and poor survival of patients. a** Analysis of GEP between MM patients with survival <2 years (n = 54) and MM patients with survival ≥4 years (n = 54) in Zhan's MM 2 dataset. Genes with significantly different mRNA expression (P < 0.05) were demonstrated. Statistical significance was determined by two-tailed Student t-test. **b** Heatmap of 24 top-ranked upregulated genes in MM patients with survival <2 years (n = 54) compared to MM patients with survival ≥ 4 years (n = 54). **c** LILRB1 expression in purified human MM cells from patients with recurrence (n = 26) and without recurrence (n = 38) in Carrasco's MM dataset. **d** LILRB1 expression in purified plasma cells from human MM (n = 12), MGUS (n = 44), or normal healthy donors (n = 22) in Zhan's MM 3 dataset. **e** LILRB1 expression in purified human MM cells of patients with stage I (n = 122), stage II (n = 88), or stage III MM (n = 83) in Broyl's MM dataset. **f** LILRB1 expression in purified MM cells of patients with low-risk MM (n = 465), defined as without del (17p), del (1p), gain (1q), t (4;14), t (14;16) and t (14;20), and patients with t (4;14) translocation (n = 87) from MMRF dataset. For (**c–f**), statistical significance was determined by two-tailed Student t-test; data are presented as mean ± SD. **g, h** Survival of MM patients with high LILRB1 (LILRB1^high) and low LILRB1 (LILRB1^low) expression in Mulligan's MM dataset (**g**) and Zhan's MM 2 dataset (**h**). MM patients were sorted by the expression level of LILRB1 and the top 25% patients with highest expression of LILRB1 were defined as LILRB1^high and the rest were defined as LILRB1^low patients. For (**g, h**), statistical significance was determined by Log-rank (Mantel-Cox) test and the p-value was demonstrated. **i** Western blot showing the protein expression of LILRB1 in human B cells from healthy donors and MM cell lines. The independent experiment was repeated three times and the representative images are shown. **j** Flow cytometry showing the expression of LILRB1 on primary MM cells from patients (n = 24). n, biological repeats, different patient samples. Source data are provided as a Source Data file.

## LILRB1 protects MM cells from lipid peroxidation-induced ferroptosis

To uncover the mechanisms underlying LILRB1's contribution to MM progression in vivo, we analyzed the RNAseq data of CTR-KD and LILRB1-KD MM cells sorted out from BM of MM-bearing mice. Ingenuity pathway analysis (IPA) of the data showed that, among the most significantly changed pathways, oxidative phosphorylation and fatty acid β-oxidation pathways were upregulated while sirtuin and semaphorin neuronal repulsive signaling pathways were downregulated in LILRB1-KD cells (Fig. 2n). Oxidative phosphorylation and β-oxidation of fatty acid pathways are essential for energy metabolism while they produce reactive oxygen species (ROS) and cause ROS-related oxidative damage in cells[49–51]. The sirtuin pathway protects cells from oxidative stress[52], and semaphorin family members are required for the

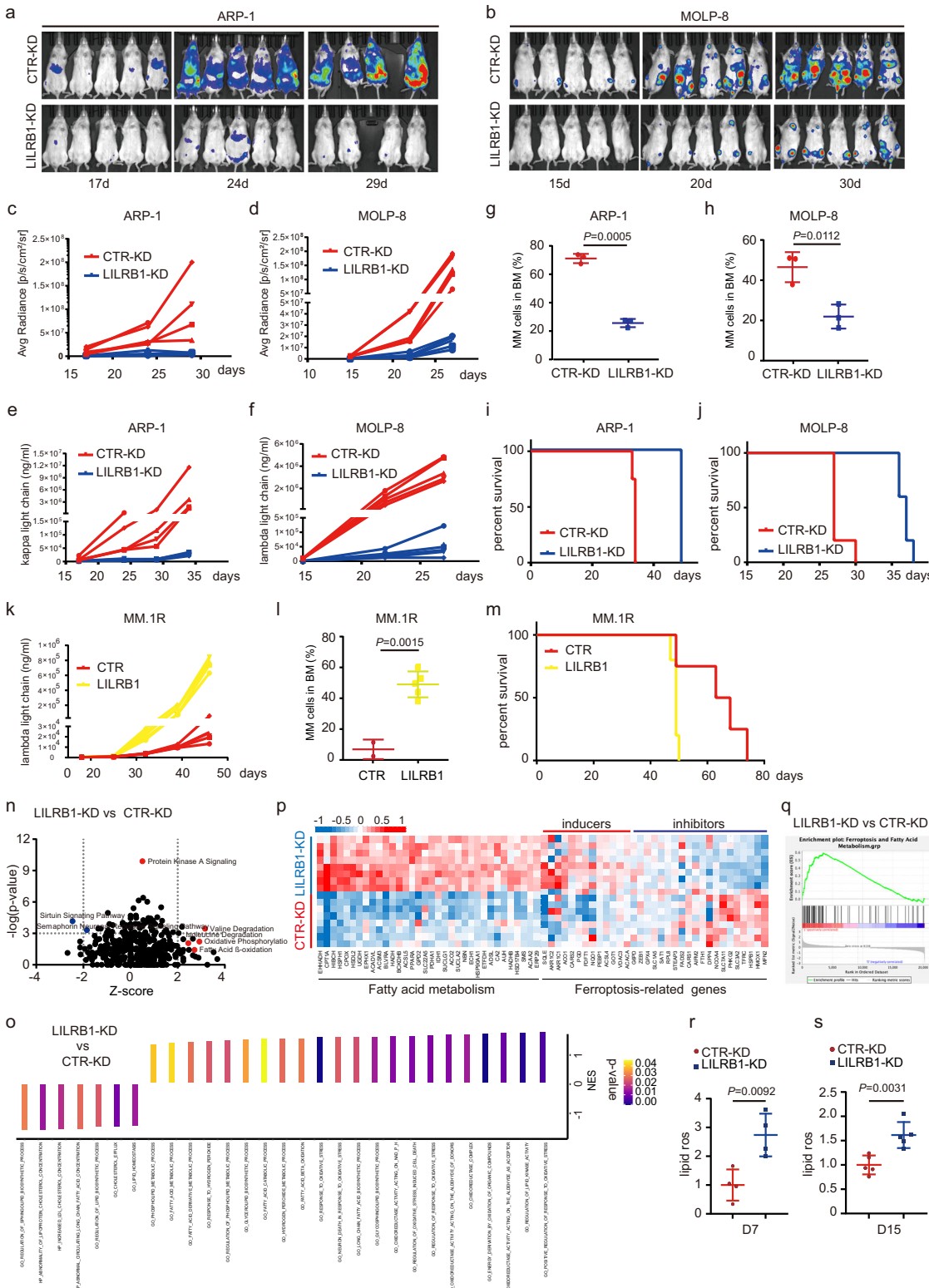

induction of antioxidant molecules through the activation of PI3K-AKT-mTOR signaling[53]. Moreover, gene set enrichment analysis (GSEA) focusing on metabolism-associated pathways showed that, compared to CTR-KD cells, LILRB1-KD cells displayed enhanced gene expression patterns associated with oxidative stress response, lipid metabolic processes, reduced lipid homeostasis, and decreased LDL-cholesterol concentration (Fig. 2o). Therefore, these results suggested that LILRB1 deficiency in MM cells leads to higher oxidative stress, ROS production, and more oxidative stress-induced cell death.

Furthermore, LILRB1-KD MM cells had a higher expression of genes associated with fatty acid metabolism and ferroptosis activation, as well as a lower expression of genes related to ferroptosis inhibition compared to CTR-KD MM cells (Fig. 2p). Fatty acid metabolism and ferroptosis gene sets were enriched in LILRB1-KD MM cells (Fig. 2q). Heme metabolism-related genes, which are important for iron metabolism and ferroptosis[54–56], were also highly enriched in LILRB1-KD MM cells (Supplementary Fig. 2d). Thus our data suggested LILRB1 deficiency in MM cells activated ferroptosis-related pathways.

**Fig. 2 | LILRB1 deficiency slows MM development and reduces tumor burden with activation of ferroptosis in vivo. a**–**j** NSG mice were injected with $2 \times 10^6$ CTR-KD or LILRB1-KD ARP-1 or MOLP-8 cells (ARP-1, male mice; MOLP-8, female mice). After 2-3 weeks of injection, MM-bearing mice were followed by bioluminescent imaging for tumor burden weekly (**a, b**) and summarized results of bioluminescent signals were shown (**c, d**). **e, f** Tumor burden measured as serum concentration of κ light chain or λ light chain in CTR-KD or LILRB1-KD MM-bearing mice. For (**a**–**f**), $n = 5$. **g, h** Tumor burden was measured as percentage of CD138⁺ MM cells in the BM in CTR-KD or LILRB1-KD MM-bearing mice ($n = 3$). Data are presented as mean ± SD. **i, j** Survival curve of CTR-KD or LILRB1-KD MM-bearing mice (**i**, $n = 4$; **j**, $n = 5$). **k**–**m** NSG mice (female) were injected with $2 \times 10^6$ CTR- or LILRB1-overexpressing MM.1 R cells, followed by monitoring of tumor burden and survival. **k** Tumor burden measured as serum concentration of λ light chain in CTR- or LILRB1-overexpressing MM.1R-bearing mice ($n = 5$). **l** Tumor burden measured as percentage of CD138⁺ MM.1 R cells in the BM in CTR- or LILRB1-overexpressing MM.1R-bearing mice (CTR, $n = 2$; LILRB1, $n = 5$). Data are presented as mean ± SD. **m** Survival curve of CTR- or LILRB1-overexpressing MM.1 R -bearing mice (CTR, $n = 4$; LILRB1, $n = 5$). **n, o** Pathway analysis of changes in RNAseq data between CTR-KD and LILRB1-KD ARP-1 cells sorted from the bone marrow of MM-bearing NSG mice. NES, normalized enrichment score. Heat map (**p**) and gene set enrichment analysis (**q**) of fatty acid metabolism- and ferroptosis-related genes in RNAseq data of CTR-KD and LILRB1-KD ARP-1 cells sorted from the bone marrow of MM-bearing NSG mice. **r, s** NSG mice were injected with $5 \times 10^6$ CTR-KD or LILRB1-KD ARP-1 cells ($n = 9$). After 7 days (**r**, $n = 4$) or 15 days (**s**, $n = 5$) of injection, lipid peroxidation of MM cells in the BM was detected. $n$, biological repeats, different mice samples. Data are presented as mean ± SD. Statistical significance was determined by two-tailed Student $t$-test. Source data are provided as a Source Data file.

To determine whether LILRB1-KD MM cells had more lipid peroxidation-induced ferroptosis in vivo, we detected the levels of lipid ROS in MM cells in the BM of tumor-bearing mice on days 7 and 15 after MM inoculation. We found that CTR-KD MM cells had fewer lipid ROS than LILRB1-KD MM cells (Fig. 2r, s), and the percentage and number of CTR-KD MM cells in the BM of tumor-bearing mice were significantly higher than those bearing LILRB1-KD MM cells (Supplementary Fig. 2e, f).

To further confirm the effect of LILRB1 deficiency on ferroptosis, we used ferroptosis inducers RSL3[19,57], Fin56[58,59], or erastin[19,60] to treat MM cells in vitro. RSL3, Fin56, or erastin induced higher lipid peroxidation (Fig. 3a, b; Supplementary Fig. 3a–e) and more ferroptotic cell death (Fig. 3c, d; Supplementary Fig. 3f) in LILRB1-KD MM cells compared to CTR-KD MM cells. We also treated MM cells with AA, which is identified in the supernatant of BM aspirates and reported to induce ferroptotic cell death in MM cells[22–24]. AA-treated LILRB1-KD MM cells had higher lipid peroxidation (Fig. 3e; Supplementary Fig. 3g, h) and more cell death (Fig. 3f) than CTR-KD MM cells. As glutamine[28,61] is essential for the production of ferroptosis protector GSH[62,63], we depleted glutamine from the culture medium of MM cells and observed that higher levels of lipid ROS were induced in MM cells (Supplementary Fig. 3i). Furthermore, LILRB1-KD MM cells displayed more cell death than CTR-KD MM cells after glutamine depletion (Supplementary Fig. 3j) and cell death induced by glutamine deprivation was dose-dependent (Supplementary Fig. 3k). In addition, lipid peroxidation (Supplementary Fig. 3l) and cell death (Supplementary Fig. 3m) induced by glutamine deprivation could be inhibited by ferroptosis inhibitors ferrostatin-1 (Fer-1) or deferoxamine (DFO)[64], indicating that glutamine deprivation-induced cell death was ferroptotic cell death. Furthermore, to confirm whether the inhibition of MM progression in LILRB1-KD MM-bearing mice was dependent on ferroptosis induced by the deficiency of LILRB1, we treated luciferase-expressing CTR-KD or LILRB1-KD ARP-1-bearing mice with ferroptosis inhibitor liproxstatin-1[65]. Treatment of liproxstatin-1 reversed the inhibition of MM progression induced by LILRB1 deficiency in vivo (Fig. 3g–i), indicating that LILRB1 indeed promotes MM progression in vivo by inhibiting MM cell ferroptosis. Additionally, treatment with RSL3 in NSG mice did not significantly affect the tumor burden of CTR-KD ARP-1 cells; however, it inhibited MM progression in mice bearing LILRB1-KD ARP-1 cells (Supplementary Fig. 3n–p). Consistently, LILRB1-overexpressing MM.1 R cells demonstrated lower lipid peroxidation and fewer ferroptotic cell death induced by RSL3, Fin56, and AA compared to CTR MM.1 R cells (Fig. 3j–m; Supplementary Fig. 3q–s).

Primary MM patient samples were utilized to investigate the correlation between LILRB1 expression and ferroptosis. Our findings revealed that MM samples expressing LILRB1 exhibited lower lipid ROS levels compared to those lacking LILRB1 expression (Supplementary Fig. 3t). Moreover, in primary MM samples containing subsets of cells with varying levels of LILRB1 expression, we observed that MM cells with lower LILRB1 expression displayed higher lipid ROS levels compared to those with higher LILRB1 expression (Fig. 3n), which is consistent with our cell line data. Additionally, while LILRB1-negative patient samples demonstrated increased cell death when treated with RSL3 (Supplementary Fig. 3u), most LILRB1-positive patient samples exhibited minimal or no response to RSL3 treatment (Supplementary Fig. 3v). Furthermore, analysis of MM patient datasets (Zhan MM datasets[32]) was conducted to compare the expression profiles of ferroptosis-related genes in MM patients. Our analysis revealed that MM patients with higher LILRB1 expression had an enrichment of genes associated with the negative regulation of ferroptosis (Fig. 3o).

Taken together, these data support that LILRB1 protects MM cells from lipid peroxidation-induced ferroptosis.

## LILRB1 interacts with LDLRAP1 and LDLR to facilitate LDL/cholesterol uptake

Considering that B2M is a ligand of LILRB1[6], we explored whether B2M may play a role in LILRB1-mediated ferroptosis in MM cells. KD of B2M or use of anti-LILRB1 antibody had no significant impact on ferroptotic cell death (Supplementary Fig. 4a–c), and anti-LILRB1 antibody did not inhibit the colony-formation ability of MM cells (Supplementary Fig. 4d), indicating that LILRB1 may interact with other novel membrane target proteins for its function in lipid peroxidation. We then performed a co-immunoprecipitation (co-IP) assay with anti-LILRB1 or anti-IgG antibodies using cell lysates of ARP-1 cells (Supplementary Fig. 4e, f) and sent captured proteins for liquid chromatography with tandem mass spectrometry (LC-MS/MS) analysis. Overall, 300 potential LILRB1-binding proteins were identified (Supplementary Data 2), which were categorized according to their biological functions (Fig. 4a). Since our GEP data demonstrated that LILRB1 affected metabolic pathways including fatty acid and LDL-cholesterol metabolisms (Fig. 2o), we selected proteins involved in cholesterol transport and LDL particle clearance and identified low-density lipoprotein receptor adapter protein 1 (LDLRAP1) as a potential target (Fig. 4b). We performed a co-IP assay using 293 T cells that transiently expressed exogenous (Fig. 4c) and ARP-1 cells that express endogenous (Fig. 4d) LDLRAP1 and LILRB1 proteins and confirmed that LILRB1 did bind with LDLRAP1, in line with the LC-MS/MS data.

As LDLRAP1 is a cytosolic adapter protein interacting with the cytoplasmic tail of LDLR and mediates the uptake of LDL through endocytosis of the LDL–LDLR complex for exogenous cholesterol delivery[66], we performed a co-IP assay to determine whether LILRB1 interacts with LDLR. Our results showed that LDLR and LILRB1 interacted with each other in both exogenous (Fig. 4e) and endogenous systems (Fig. 4f). Moreover, immunofluorescence (IF) staining demonstrated that LILRB1, LDLRAP1, and LDLR were co-localized in the membrane with each other (Fig. 4g–i), supporting our hypothesis that these three proteins form a complex in the membrane. To further investigate the binding regions of LILRB1 with LDLR or LDLRAP1, we generated various truncated LILRB1 plasmids (Supplementary Fig. 4g).

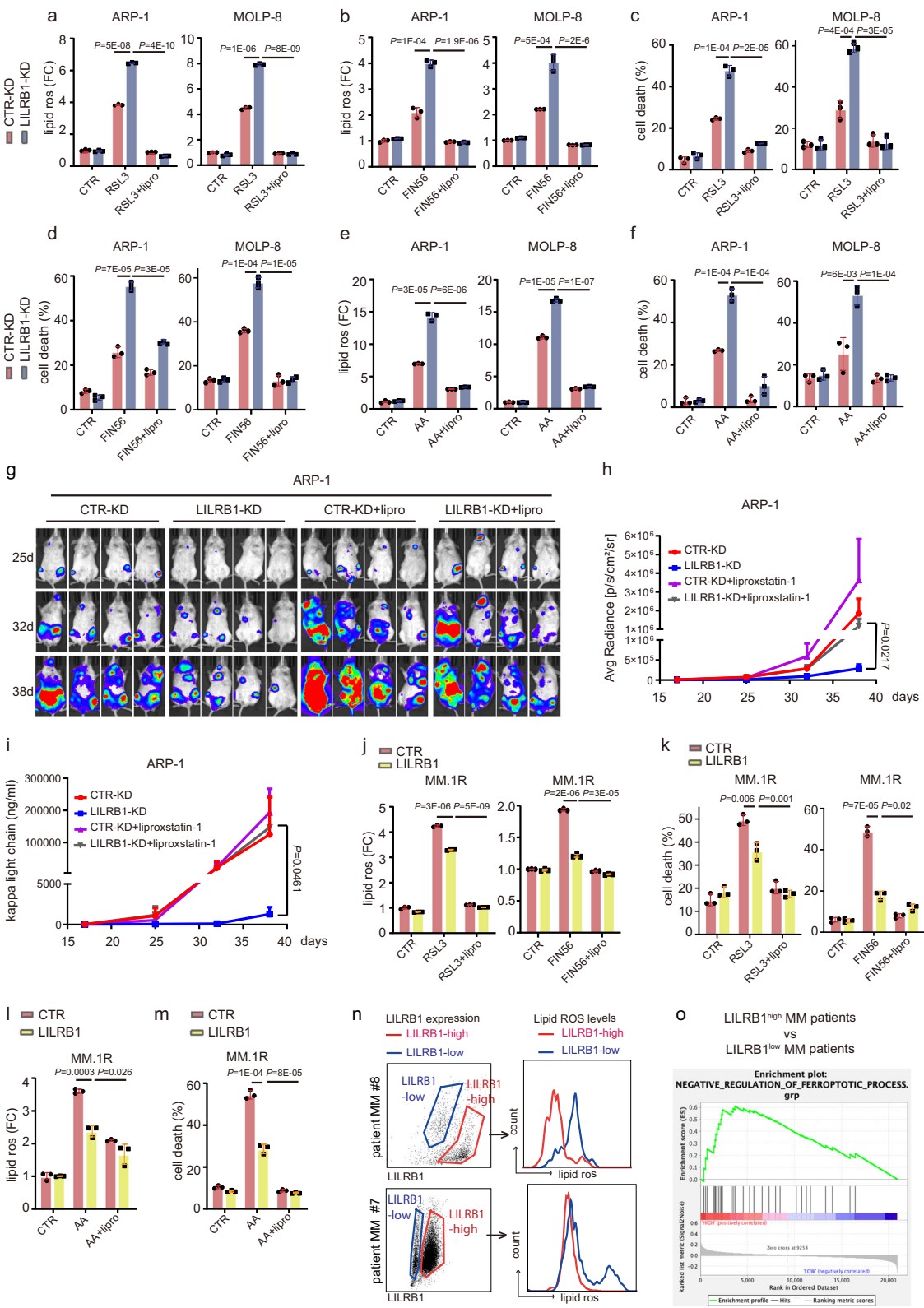

Our immunoprecipitation results showed that LILRB1-A4 (1-495aa), -A5 (1-540aa), -A6 (1-569aa), -A7 (1-621aa), and -A8 (1-651aa) interacted with both LDLRAP1 and LDLR (Supplementary Fig. 4h, i), suggesting that the binding region may be located within the 1-495aa segment. While the interaction between LILRB1-A3 (1-460aa), -A2 (1-312aa) and -A1 (1-221aa), and LDLRAP1 was considerably weaker compared to LILRB1-A4 and full-length LILRB1 (Supplementary Fig. 4j), interaction between LILRB1-B1 (313-661aa), -B2 (462-661aa), and -B3 (483-661aa) did not

significantly differ from that of full-length LILRB1 (Supplementary Fig. 4k). This suggests that the cytoplasmic region near the transmembrane domain (483-495aa) is crucial for LILRB1's binding with LDLRAP1. Regarding the binding with LDLR, LILRB1-A1, -B2, and B3 showed strong inhibition of interaction compared to other truncated LILRB1 variants (Supplementary Fig. 4l, m), indicating that the extracellular domain, particularly the 221-460aa region of LILRB1, is a potential binding domain with LDLR.

**Fig. 3 | LILRB1 deficiency enhances MM cell susceptibility to ferroptosis.**
**a–f** CTR-KD or LILRB1-KD MM cells were treated with ferroptosis inducer RSL3 (400 nM, 4 h-lipid peroxidation, 10 h-cell death)/FIN56 (15 μM, 18-lipid peroxidation, 24 h- cell death)/ AA (75 μM, 4 h-lipid peroxidation, 10 h-cell death) and ferroptosis inhibitor liproxstatin-1 (lipro, 1 μM). Lipid peroxidation (**a**, **b**, **e**) and cell death (**c**, **d**, and **f**) of CTR-KD or LILRB1-KD MM cells was measured by flow cytometry. **g–i** NSG mice (female) were injected with 2 × 10⁶ CTR-KD or LILRB1-KD ARP-1 cells through tail vein, followed by administration of vehicle (veh) or ferroptosis inhibitor liproxstatin-1 (lipro, 15 mg/kg, ip) every day and monitoring of tumor burden (CTR-KD, n = 4; LILRB1-KD, n = 4; CTR-KD+liproxstatin-1, n = 7; LILRB1-KD +liproxstatin-1, n = 7). Representative bioluminescent imaging for tumor burden (**g**) and summarized quantification of bioluminescent imaging (mean ± SE) (**h**) are shown. **i** Tumor burden was measured as serum concentration of κ light chain and shown as mean ± SE. **j–m** CTR- or LILRB1-overexpression MM.1 R cells were treated with ferroptosis inducer RSL3 (400 nM, 4 h-lipid peroxidation, 10 h-cell death)/ FIN56 (15 μM, 18-lipid peroxidation, 24 h- cell death)/ AA (75 μM, 4 h-lipid peroxidation, 10 h-cell death) and liproxstatin-1 (lipro, 1 μM). Lipid peroxidation (**j**, **l**) and cell death (**k**, **m**) of CTR-KD or LILRB1-KD MM cells was measured by flow cytometry. **n** Lipid peroxidation of primary MM patient samples containing subsets of MM cells with different LILRB1 expression was measured. **o** Gene set enrichment analysis of negative regulation of ferroptosis process genes between LILRB1 high expression MM patients and LILRB1 low expression MM patients in Zhan's MM 2 dataset. MM patients were sorted by the expression level of LILRB1: the top 100 patients with highest expression of LILRB1 were defined as LILRB1 high expression and the bottom 100 patients with lowest expression were defined as LILRB1 low expression. For (**g–i**), n, biological repeats, different mice samples. For (**a–f**, **j–m**), n, biological repeats, independent experimental samples; data are summary of three independent experimental samples and shown as mean ± SD. Statistical significance was determined by two-tailed Student t-test. Source data are provided as a Source Data file.

Based on the above findings, we speculated that LILRB1 may function in the process of LDL uptake. To verify this, chemically modified LDL labeled with red fluorescence was used. IF staining revealed a co-localization of LDL and LILRB1 (Fig. 4j). We then compared LDL uptake in CTR-KD and LILRB1-KD MM cells. Both flow cytometry (Fig. 4k) and IF staining (Fig. 4l) showed that LDL uptake was inhibited in LILRB1-KD MM cells. We also determined the uptake of LDL/cholesterol by detecting the changes in the extracellular cholesterol concentrations in the culture supernatant of MM cells. Our results illustrated that LILRB1-KD cells exhibited a reduced cholesterol uptake compared to CTR-KD cells (Fig. 4m). Simultaneously, intracellular cholesterol did not show a significant difference between CTR-KD cells and LILRB1-KD MM cells (Fig. 4n), suggesting that LILRB1-KD cells may activate the synthesis pathway to maintain cholesterol balance. Conversely, LILRB1-overexpressing MM cells exhibited an increased LDL/cholesterol uptake (Fig. 4o), without significant difference in intracellular cholesterol compared to CTR MM cells (Fig. 4p). We also determined whether LILRB1 could interact with apolipoprotein B (APOB) or apolipoprotein E (APOE), which are components of LDL and ligand of LDLR. The results showed that LILRB1 did not bind with APOE or APOB (Supplementary Fig. 4n, o). Taken together, these data demonstrated that LILRB1 interacts with LDLR and LDLRAP1 to form a complex to facilitate LDL uptake in MM cells.

### Cholesterol/LDL protects MM cells from ferroptotic cell death
We then went on to investigate whether LDL or cholesterol affects the induction of ferroptosis in MM cells. Interestingly, MM cell death induced by ferroptosis inducer RSL3 (Fig. 5a; Supplementary Fig. 5a) or FIN56 (Fig. 5b; Supplementary Fig. 5b) was significantly inhibited by LDL in a dose-dependent manner. As LDL is a group of particles responsible for cholesterol delivery[67,68], cholesterol may be responsible for LDL-mediated protection of MM cell ferroptosis. Indeed, similar to LDL, cholesterol reduced the cell death induced by ferroptosis inducers (Fig. 5c, d; Supplementary Fig. 5c, d). Moreover, LDL (Fig. 5e, f; Supplementary Fig. 5e, f) and cholesterol (Fig. 5g, h; Supplementary Fig. 5g, h) reduced cell death induced by ferroptosis inducers in both CTR-KD and LILRB1-KD MM cells. Similar to the results of ferroptosis inducers, LDL/cholesterol could inhibit AA-induced ferroptotic cell death (Fig. 5i, j; Supplementary Fig. 5i, j) and LILRB1 deficiency increased cell death under AA treatment (Fig. 5k, l and Supplementary Fig. 5k, l). Furthermore, we examined the effects of LDL on CTR-KD and LILRB1-KD cells in NSG mice. Our results revealed that LDL treatment increased tumor burden of MM-bearing mice. While LILRB1-KD inhibited MM cell progression in vivo, LDL treatment significantly promoted MM progression in both LILRB1-KD and CTR-KD MM-bearing mice (Supplementary Fig. 5m−o), which aligns well with our in vitro data. These data indicated that the uptake of cholesterol/LDL may be involved in the protective effect of LILRB1 against the induction of ferroptosis in MM cells.

### LILRB1 promotes LDL uptake by enhancing the interaction with LDLR and LDLRAP1
Next, the question arises: how does LILRB1 facilitate LDL uptake? Based on our findings that LILRB1 formed a complex with LDLR and LDLRAP1 (Fig. 4), we wondered whether LILRB1 enhances the interaction between LDLR and LDLRAP1 and whether KD of LILRB1 abolishes the formation of the complex and reduces LDL uptake. Indeed, co-IP results showed that the interaction between LDLR and LDLRAP1 was much weaker in LILRB1-KD MM cells compared to CTR-KD MM cells (Fig. 6a, b). We also knocked down LDLR or LDLRAP1 to determine whether LDLR or LDLRAP1 deficiency has similar effects as LILRB1 KD. As expected, KD of LDLR or LDLRAP1 also inhibited LDL uptake (Fig. 6c–f; Supplementary Fig. 6a) and increased RSL3-induced ferroptotic cell death (Fig. 6g, h; Supplementary Fig. 6b, c). Moreover, analysis of MM patient datasets showed that patients with higher expression of LDLR and LDLRAP1 had inferior survival (Fig. 6i, j; Supplementary Fig. 6d, e). In addition, analysis of the Multiple Myeloma Research Foundation (MMRF) datasets showed that MM patients with hypercholesterolemia exhibited a lower expression of pathways associated with ferroptosis activation compared to those with normal cholesterol levels (Fig. 6k). These findings align with our in vitro data and further suggest that cholesterol can protect MM cells from ferroptosis. Taken together, we revealed that LILRB1 is involved in LDL uptake by enhancing the interaction between LDLR and LDLRAP1 and thus is an important chaperone protein to maintain LDL/cholesterol homeostasis in MM cells.

### Cholesterol metabolic alteration increases SQLE and down-regulates squalene, rendering LILRB1-KD MM cell sensitivity to ferroptosis induction
As we observed that KD of LILRB1 inhibited LDL uptake, we speculated that inhibition of LDL/cholesterol uptake may lead to the activation of the cholesterol synthesis pathway as a feedback compensatory regulation and upregulate the expression of squalene monooxygenase (SQLE), a rate-limiting enzyme in the cholesterol synthesis pathway catalyzing the oxidation of squalene. As expected, KD of LDLR or LDLRAP1 increased the expression of SQLE (Fig. 6c, d). We then detected the expression of SQLE in CTR-KD and LILRB1-KD MM cells, and observed that, in LILRB1-KD MM cells, protein levels of SQLE were upregulated, which could be reversed by cholesterol or LDL (Fig. 7a, b). Moreover, ferroptosis inducer FIN56 also upregulated the expression of SQLE, which was further increased by LILRB1 deficiency (Supplementary Fig. 7a). HMGCR, another rate-limiting enzyme in the mevalonate pathway, was also detected in CTR-KD and LILRB1-KD MM cells. The expression of HMGCR in CTR-KD and LILRB1-KD MM cells was relatively low and did not show obvious difference (Supplementary Fig. 7b), while the expression of SQLE was significantly increased in LILRB1-KD MM cells.

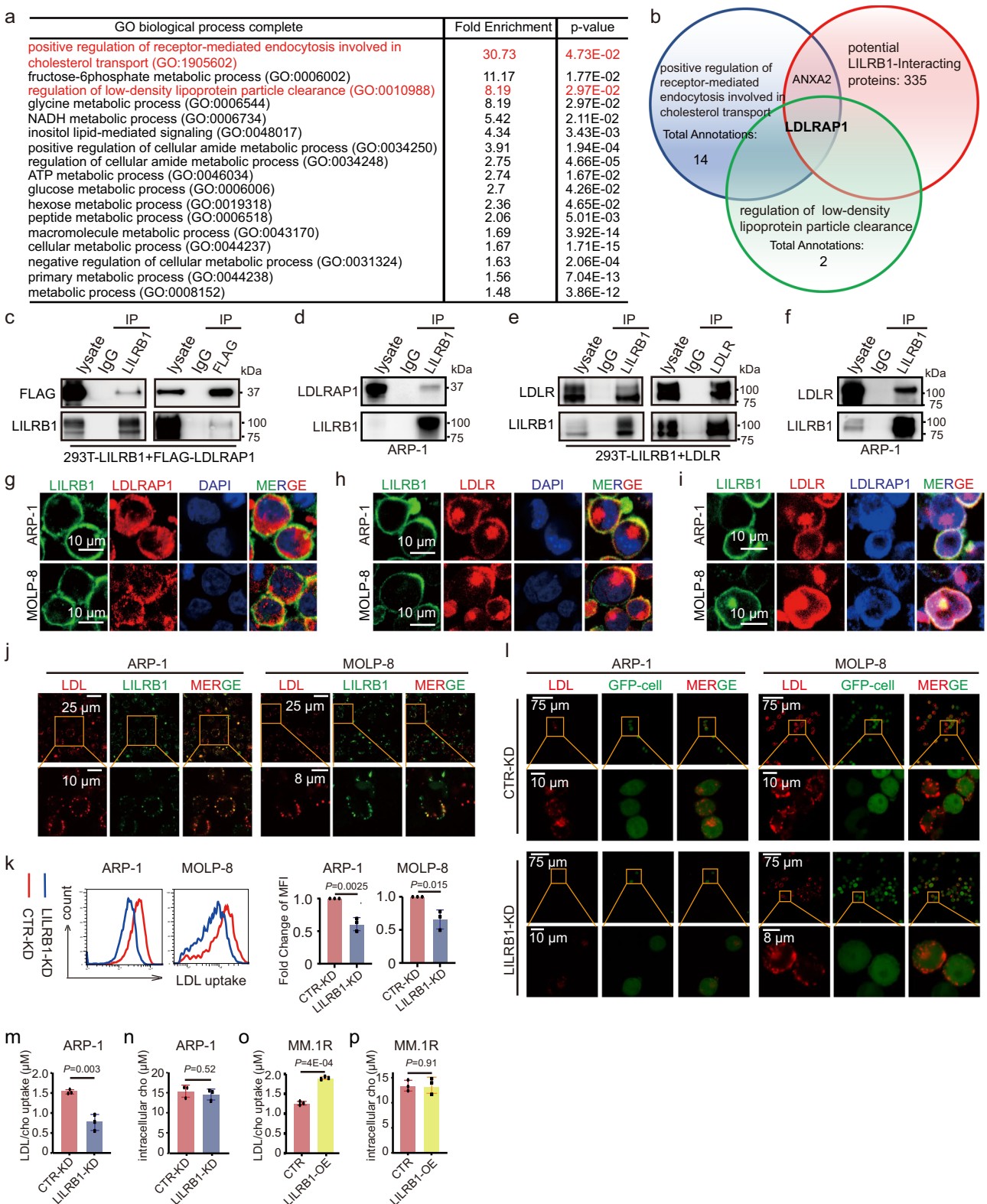

To determine which intermediates in the de novo cholesterol synthesis are important for ferroptosis, selective inhibitors targeting the different steps of the cholesterol synthesis pathway were used (Fig. 7c). Simvastatin, the 3-hydroxy-3-methylglutaryl-CoA reductase (HMGCR) enzyme inhibitor[69], increased ferroptotic cell death induced by RSL3 in both CTR-KD and LILRB1-KD MM cells (Fig. 7d; Supplementary Fig. 7c), while SQLE inhibitors such as terbinafine HCl[70] (Fig. 7e; Supplementary Fig. 7d) and NB598[71] (Fig. 7f; Supplementary

Fig. 7e), which cause the accumulation of squalene, strongly inhibited cell death induced by RSL3 in LILRB1-KD cells, indicating that LILRB1-KD MM cells may have become more sensitive to ferroptosis induction with less squalene. We then knocked down SQLE in ARP-1 cells and found that it protected MM cells from RSL3-induced cell death, suggesting a role of squalene accumulation in safeguarding against ferroptosis (Supplementary Fig. 7f–h). Furthermore, we assessed the expression of SQLE across various MM cell lines (Supplementary

**Fig. 4 | LILRB1 plays an important role in LDL uptake. a, b** Products of immunoprecipitation by anti-IgG and anti-LILRB1 antibodies were sent for mass spectrum analysis. Potential LILRB1 interacting proteins detected by MS were analyzed with Gene Ontology (GO) enrichment analysis. Enriched Pathways involved in metabolisms are shown. **c** Immunoprecipitation of LILRB1 (left panel) and FLAG-LDLRAP1 (right panel) in 293 T cells overexpressing LILRB1 and FLAG-LDLRAP1 showing the binding of LILRB1 and FLAG-LDLRAP1. **d, f** Immunoprecipitation of endogenously expressed LILRB1 in ARP-1 cells showing the binding of LILRB1 and LDLRAP1 (**d**) and LILRB1 and LDLR (**f**). **e** Immunoprecipitation of LILRB1 (left panel) and LDLR (right panel) in 293 T cells overexpressing LILRB1 and LDLR showed the binding of LILRB1 and LDLR. **g–j** Representative fluorescent confocal images showing the co-localization of LILRB1 and LDLRAP1 (**g**), co-localization of LILRB1 and LDLR (**h**), co-localization of LILRB1, LDLRAP1 and LDLR (**i**), and co-localization of LILRB1 and LDL (**j**) in ARP-1 cells or MOLP-8 cells. **k, l** CTR-KD or LILRB1-KD MM cells were cultured in FBS-free medium for 24 h and then incubated with chemically modified LDL labeled with red fluorescence for 2 h. The uptake of LDL by CTR-KD or LILRB1-KD MM cells was measured by flow cytometry (**k**). Representative fluorescent confocal images showing LDL uptake ability of CTR-KD or LILRB1-KD MM cells (**l**). **m, o** Uptake of LDL/cholesterol was detected by the changes of extracellular LDL/cholesterol concentrations in the supernatant of CTR-KD and LILRB1-KD ARP-1 cells (**m**), as well as CTR and LILRB1-OE MM.1 R cells (**o**), by fluorescence microplate reader. (**n, p**) Intracellular cholesterol concentrations of CTR-KD and LILRB1-KD ARP-1 cells (**n**), as well as CTR and LILRB1-OE MM.1 R cells (**p**), were detected with fluorescence microplate reader. For (**c–j, l**), the independent experiments were repeated three times and the representative images are shown. For (**k, m–p**), $n = 3$, independent experimental repeats; data are summary of three independent experiments and shown as mean ± SD. Statistical significance was determined by two-tailed Student $t$-test. Source data are provided as a Source Data file.

Fig. 7i). Interestingly, MM cell lines expressing LILRB1 exhibited reduced SQLE expression, suggesting a potential functional link between LILRB1 and SQLE. While assessing squalene and cholesterol levels in different MM cell lines, we did not discern a clear correlation among SQLE, LILRB1, squalene, and cholesterol levels, likely due to the diverse genetic backgrounds of the MM cell lines analyzed (Supplementary Fig. 7j–k). Then we determined squalene levels in CTR-KD and LILRB1-KD MM cells. Consistent with the results that LILRB1-KD MM cells had a higher expression of SQLE, lower squalene levels were detected in LILRB1-KD MM cells (Fig. 7g). We also evaluated Coenzyme Q10 (CoQ10), which is an antioxidant metabolite involved in the mevalonate pathway and was reported to function in the inhibitory effect on ferroptosis induced by downregulation of SQLE[72]. Our results showed a reduced CoQ10 level in LILRB1-KD cells compared to CTR-KD cells (Supplementary Fig. 7l), reinforcing our finding that KD of LILRB1 promotes ferroptosis by upregulating SQLE. Furthermore, isotope tracing analysis of squalene and CoQ10 also demonstrated that the newly synthesized squalene isotopomers and CoQ10 isotopomers were decreased in LILRB1-KD ARP-1 cells (Supplementary Fig. 7m–p).

Overall, these results demonstrate that, by facilitating LDL/cholesterol uptake, LILRB1 plays an important role in maintaining cholesterol metabolic homeostasis to protect MM cells from induction of ferroptosis in tumor microenvironment (Supplementary Fig. 8).

## Discussion

In this study, we analyzed MM patient datasets and observed that high expression of LILRB1 was closely related to the aggressive behaviors of MM. Further investigation demonstrated that MM patients with high expression of LILRB1 are closely related to higher MM recurrence rates, advanced stages, and lower survival rates, indicating that LILRB1 is an important player in MM pathogenesis and thus a promising target for MM therapy. In line with these findings, LILRB1 deficiency significantly inhibited MM progression in vivo by enhancing ferroptosis in MM cells. LC-MS/MS analysis followed by co-IP demonstrated that LDLRAP1, an adapter protein that interacts with LDLR[73], bound with LILRB1 and formed a complex together with LDLR. KD of LILRB1 inhibited LDL uptake by disrupting the interaction between LDLR and LDLRAP1 and triggering the compensatory cholesterol synthesis by upregulating the expression of SQLE that converted squalene, an anti-ferroptotic metabolite, to (S)−2,3-epoxysqualene. With less squalene to protect MM cells from lipid peroxidation, MM cells were more susceptible to the induction of ferroptosis. Thus, our research revealed that LILRB1 not only plays an important role in LDL uptake but also maintains cellular metabolic balance to protect MM cells from ferroptosis. Hence, our study identifies LILRB1 as a therapeutic target for MM patients (Supplementary Fig. 8).

We discovered a function of LILRB1 in promoting LDL uptake through its interaction with LDLR and LDLRAP1. LDLR is ubiquitously expressed in almost all tissues and responsible for internalization of plasma LDL-cholesterol[74–76]. However, the major determinant of LDL-cholesterol uptake is unclear. Accordingly, our knowledge about the mechanisms regulating LDL uptake is far from complete. Here, we showed that LILRB1 promotes LDL uptake through its interaction with LDLR and LDLRAP1. Although LDL-cholesterol uptake is LDLR-dependent, the uptake rate is largely affected by LILRB1 expression to form the LDLR-LILRB1-LDLRAP1 complex in MM cells. Analysis of MM patient datasets showed that high expressions of LILRB1 in MM cells were associated with patient inferior survival, suggesting that LILRB1 regulated LDL-cholesterol uptake is important for MM pathogenesis.

As an essential component of cell membranes, cholesterol is the most abundant lipid and a substrate of lipid peroxidation. Although cholesterol hydroperoxides are reported to induce cell death[77], the role of cholesterol in tumor cell survival and ferroptosis has not been fully clarified. HDL-like nanoparticles, which could inhibit cholesterol uptake, activated a compensatory metabolic response including increased cholesterol synthesis activity, and resulted in ferroptotic cell death[78]. Consistent with these reports, our data showed that cholesterol/LDL uptake protects MM cells from ferroptosis. Moreover, we uncovered the mechanisms that in LILRB1-KD cells, inhibition of LDL/cholesterol uptake upregulated SQLE and downregulated squalene, and impaired ability to protect cells from lipid ROS. Another group also reported that loss of SQLE led to the accumulation of squalene that altered the cellular lipid profile and protected cancer cells from ferroptotic cell death[79], which is consistent with our data. Meanwhile, LDL or cholesterol treatment on various cells led to the accumulation of squalene[80], aligning with our data that LDL/cholesterol treatment protects MM cells from ferroptotic cell death. By helping with LDL uptake, LILRB1 acts as an important mediator in maintaining cholesterol metabolism homeostasis and preventing MM cells from ferroptosis. These findings thus bridge the gap between LILRB1 function and cholesterol metabolism and ferroptosis in MM cells. Additionally, a recent report found that cells cultured in lipoprotein-deficient human serum exhibited a marked reduction in RSL3-induced ferroptosis[81]. Interestingly, the addition of HDL, but not LDL or VLDL, restored the sensitivity to ferroptosis under lipoprotein deficiency[81]. These findings suggest distinct roles of HDL and LDL in ferroptosis and highlight the need for further investigation into the underlying mechanisms. The BM microenvironment interplays with MM cells. Hypocholesterolemia is one of the symptoms in MM patients, indicating an increased utilization of cholesterol by MM cells[26–28] and suggesting that MM cells are addicted to cholesterol and maintenance of cholesterol homeostasis could be especially important for MM cells. Moreover, low concentrations of cholesterol in MM patient's serum might further upregulate SQLE expression in MM cells as a compensation regulation to synthesize more cholesterol, leading to lower squalene levels and rendering MM cells to be sensitive to the stress of lipid peroxidation. Therefore, targeting LILRB1 to disrupt the cholesterol homeostasis and promote MM cell ferroptosis could be an effective strategy for MM treatment.

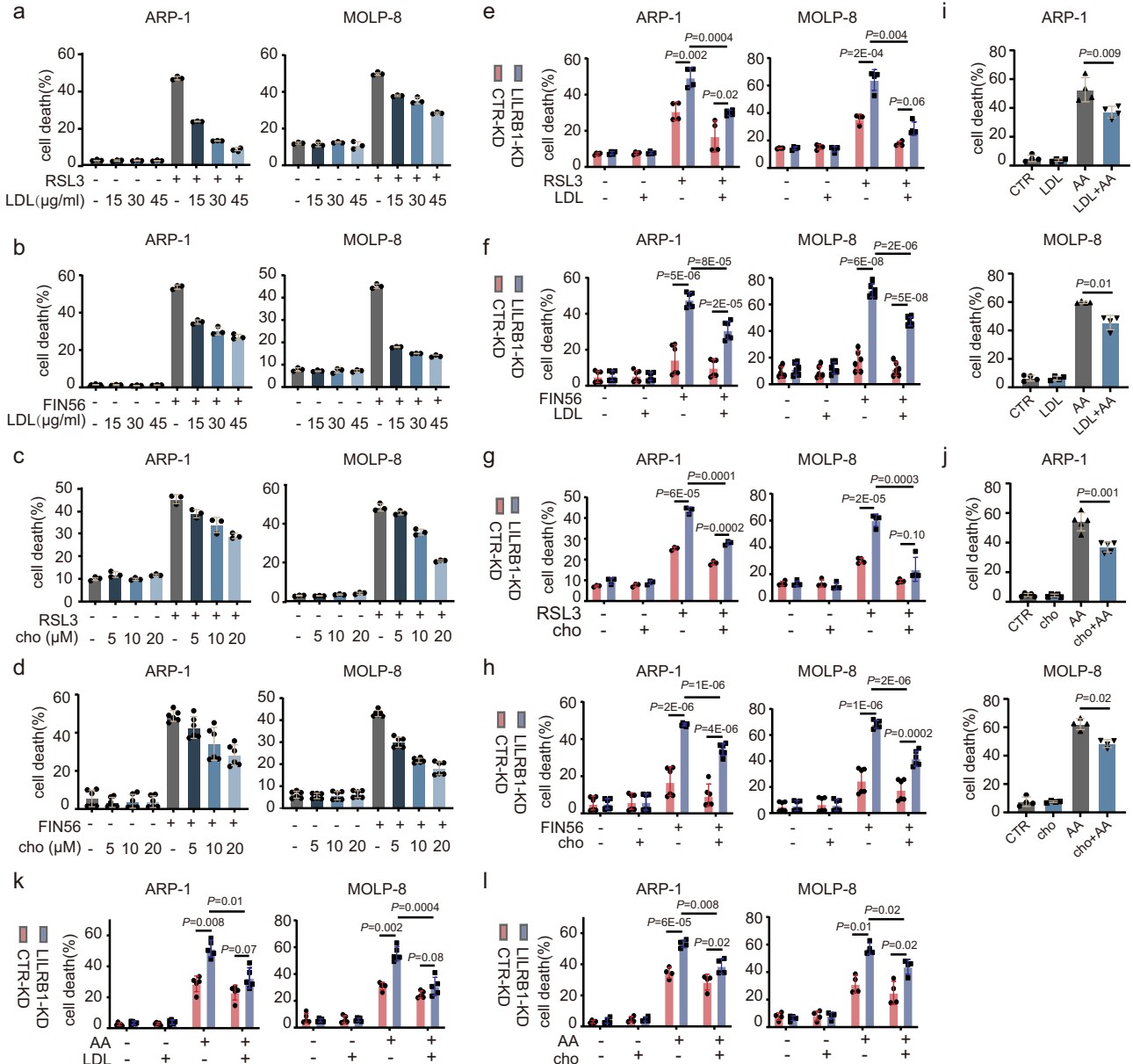

**Fig. 5 | Cholesterol/LDL protects MM cells from ferroptotic cell death.**
**a**–**l** Summarized data showing cell death of ARP-1 and MOLP-8 MM cells. Cells were incubated with LDL/cholesterol for 12 h before addition of ferroptosis inducers. **a**, **c** Cell death of MM cells treated with ferroptosis inducer RSL3 (400 nM) and different concentrations of LDL (0, 15, 30, 45 μg/ml) (**a**) or cholesterol (0, 5, 10, 20 μM) (**c**) for 10 h. **b**, **d** Cell death of MM cells treated with ferroptosis inducer FIN56 (15 μM) and different concentrations of LDL (0, 15, 30, 45 μg/ml) (**b**) or cholesterol (0, 5, 10, 20 μM) (**d**) for 24 h. **e**, **g** Cell death of CTR-KD or LILRB1-KD MM cells treated with RSL3 (400 nM) and/or LDL (30 μg/ml) (**e**) or cholesterol (10 μM)

(**g**). **f**, **h** Cell death of CTR-KD or LILRB1-KD MM cells treated with FIN56 (15 μM) and/or LDL (30 μg/ml) (**f**) or cholesterol (10 μM) (**h**). **i**, **j** Cell death of MM cells treated with AA (75 μM) and LDL (30 μg/ml) (**i**) or cholesterol (10 μM) (**j**) for 10 h. **k**, **l** Cell death of CTR-KD or LILRB1-KD MM cells treated with AA (75 μM) and/or LDL (30 μg/ml) (**k**) or cholesterol (10 μM) (**l**). Data are summary of at least three independent experiments and showed as mean ± SD. **a**–**c** g-ARP1, *n* = 3; (**d**, **f**, and **h**) *n* = 6; (**e**, **i**, and **l**) g-MOLP-8, j-MOLP-8, *n* = 4;k, j-ARP-1, *n* = 5; *n*, independent experimental repeats; Statistical significance was determined by two-tailed Student *t*-test. Source data are provided as a Source Data file.

B2M, one of the ligands of LILRB1[6], is highly expressed by MM cells and elevated in MM patients' serum[40]. Our study demonstrated that B2M did not affect LILRB1's function in protecting MM cells. Consistent with our results, another report found that blocking LILRB1 with antibodies on MM did not alter NK-92-mediated lysis[82]. Further study is needed to determine whether or which of other ligands may play a role in LILRB1's function in MM cells. Different from our observation that LILRB1 promotes the progression of MM cells, previous studies on B cells reported that LILRB1–HLA-G interaction inhibits both naive and memory B cell functions[14], and LILRB1 downregulates immunoglobulin and cytokine production by human B lymphocytes[16]. These

observations may be attributed to the potential variation of the function of LILRB1 in normal B/plasma cells versus MM cells. Additionally, distinct pathways associated with LILRB1 may play different roles in different types of cells. Interestingly, another study reported that overexpression of LILRB1 could increase the specific killing by NK and T cells in vitro[83]. It is possible that the effect of MM-derived LILRB1 in the immune responses and tumor cells may be different. Considering the potential role of immune cell-derived LILRB1 within the MM microenvironment, further investigation into the overall effects of targeting LILRB1 in MM patients is needed. Given that LILRB1 blockade on T cells and NK cells has been shown to enhance

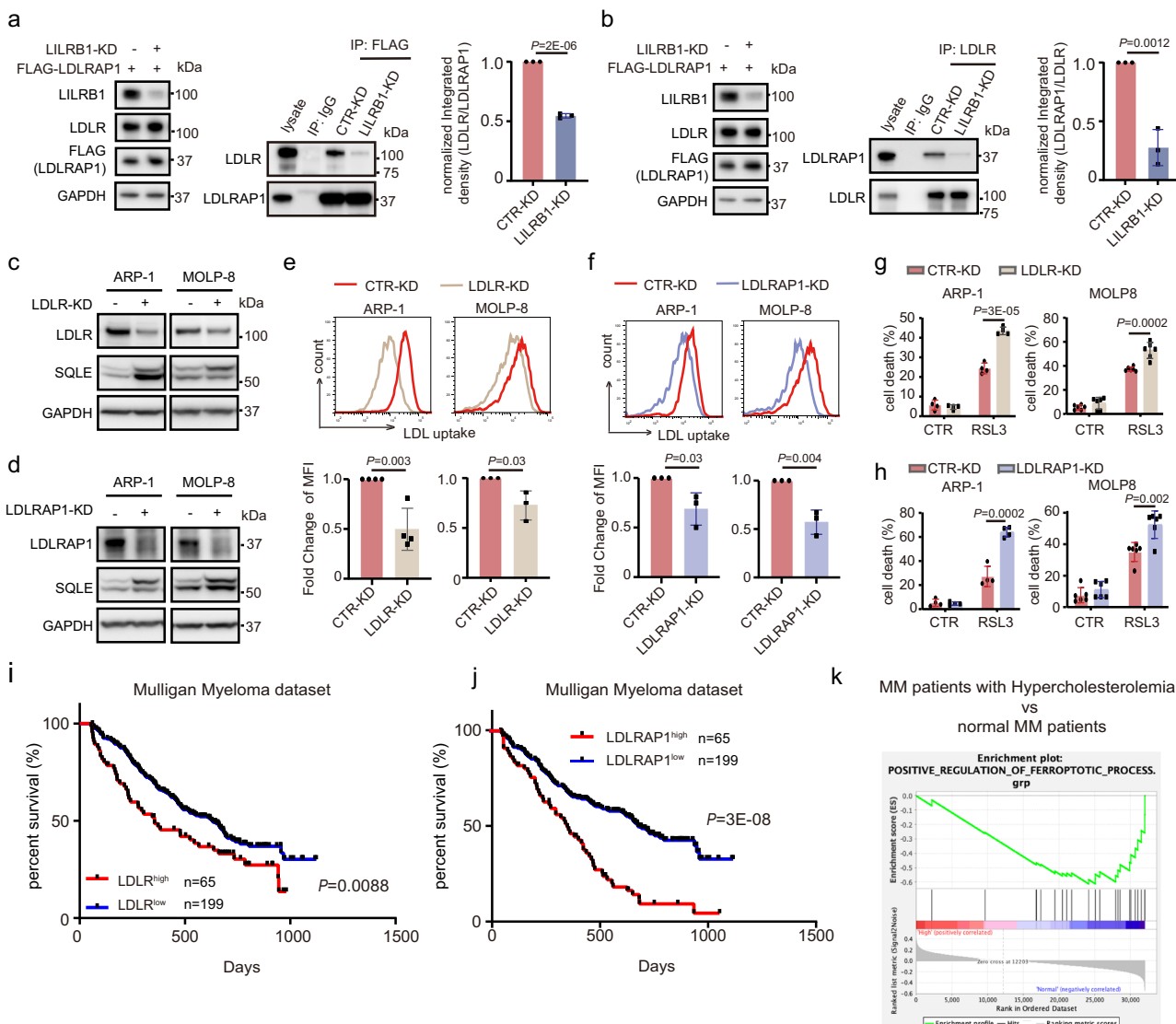

**Fig. 6 | LILRB1 promotes LDL uptake by enhancing the interaction between LDLR and LDLRAP1. a–b** Immunoprecipitation of FLAG-LDLRAP1 (**a**) or LDLR (**b**) showing the interaction between LDLR and FLAG-LDLRAP1 in CTR-KD or LILRB1-KD ARP-1 cells. The left panel shows the protein levels in loading lysates and the middle panel shows the results of immunoprecipitation. Quantification of the relative band density of LDLR/LDLRAP1 (**a**) or LDLRAP1/LDLR (**b**) in the immunoprecipitation is depicted in the right panel. **c**, **e**, and **g** CTR-KD or LDLR-KD MM cells were determined the expression of LDLR, SQLE and GAPDH by western blot (**c**), LDL uptake ability by flow (**e**), and sensitivity to ferroptosis inducer RSL3 (400 nm, 10 h) (**g**). **d**, **f**, **h** CTR-KD or LDLRAP1-KD MM cells were determined the expression of LDLRAP1, SQLE and GAPDH by western blot (**d**), LDL uptake ability by flow (**f**), and sensitivity to ferroptosis inducer RSL3 (400 nm, 10 h) (**h**). For (**a**, **b**, and **e–h**), data are presented as mean ± SD; statistical significance was determined by two-tailed

Student *t*-test. **a**, **b**, and **e**-MOLP-8, **f**, n = 3; (**g**, **h**)-ARP-1, n = 4; (**g**, **h**)-MOLP-8, n = 6; n, independent experimental repeats; For (**a–d**), the independent experiments were repeated three times and the representative images are shown. **i**, **j** Survival of MM patients with high LDLR (LDLRhigh) and low LDLR (LDLRlow) expression (**i**) or high LDLRAP1 (LDLRAP1high) and low LDLRAP1 (LDLRAP1low) expression (**j**) in Mulligan's MM dataset. MM patients were sorted by the expression level of LDLR/LDLRAP1 and the top 25% patients with highest expression of LDLR/LDLRAP1 were defined as LDLRhigh/LDLRAP1high and the rest were defined as LDLRlow/LDLRAP1low patients. For (**i**, **j**), Statistical significance was determined by Log-rank (Mantel-Cox) test and p-values are shown. **k** Gene set enrichment analysis of positive regulation of ferroptosis process genes between MM patients with hypercholesterolemia and normal MM patients in MMRF dataset. Source data are provided as a Source Data file.

their tumor-killing effects[9,84], it is plausible that targeting LILRB1 in MM patients could elicit responses from both MM cells and immune cells. Thus, our research on LILRB1's function in MM cells provides essential data supporting LILRB1 as a promising therapeutic target for MM patients.

To translate our findings to the clinic, ways to target LILRB1 in MM patients need to be determined. Antagonistic LILRB1 mAbs have been reported to enhance the antitumor functions of NK cells in several types of tumors[9]. However, these mAbs may not be suitable for disrupting the interaction of LILRB1 with LDLRAP1, which is a cytosolic protein[85]. Small molecule inhibitors of LILRB1 have not been developed

based on the literature. The design of small molecules that disrupt the interaction between LILRB1, LDLR, and LDLRAP1 may be a promising approach and more research needs be performed. We are conducting investigations on proteolysis-targeting chimeric molecules (PROTACs)[86], consisting of LILRB1 ligands linked with recruiting elements for membrane-associated RING-type E3 ligase, to hijack the ubiquitin-proteasome system in cells to degrade LILRB1. These PRO-TACs may counteract LILRB1-mediated immune suppression in immune cells while enhancing ferroptotic cell death in MM cells. For instance, CD8+ T cells are known to promote tumor cell ferroptosis through IFNγ secretion[65,87]. However, LILRB1 in CD8+ T cells acts as a

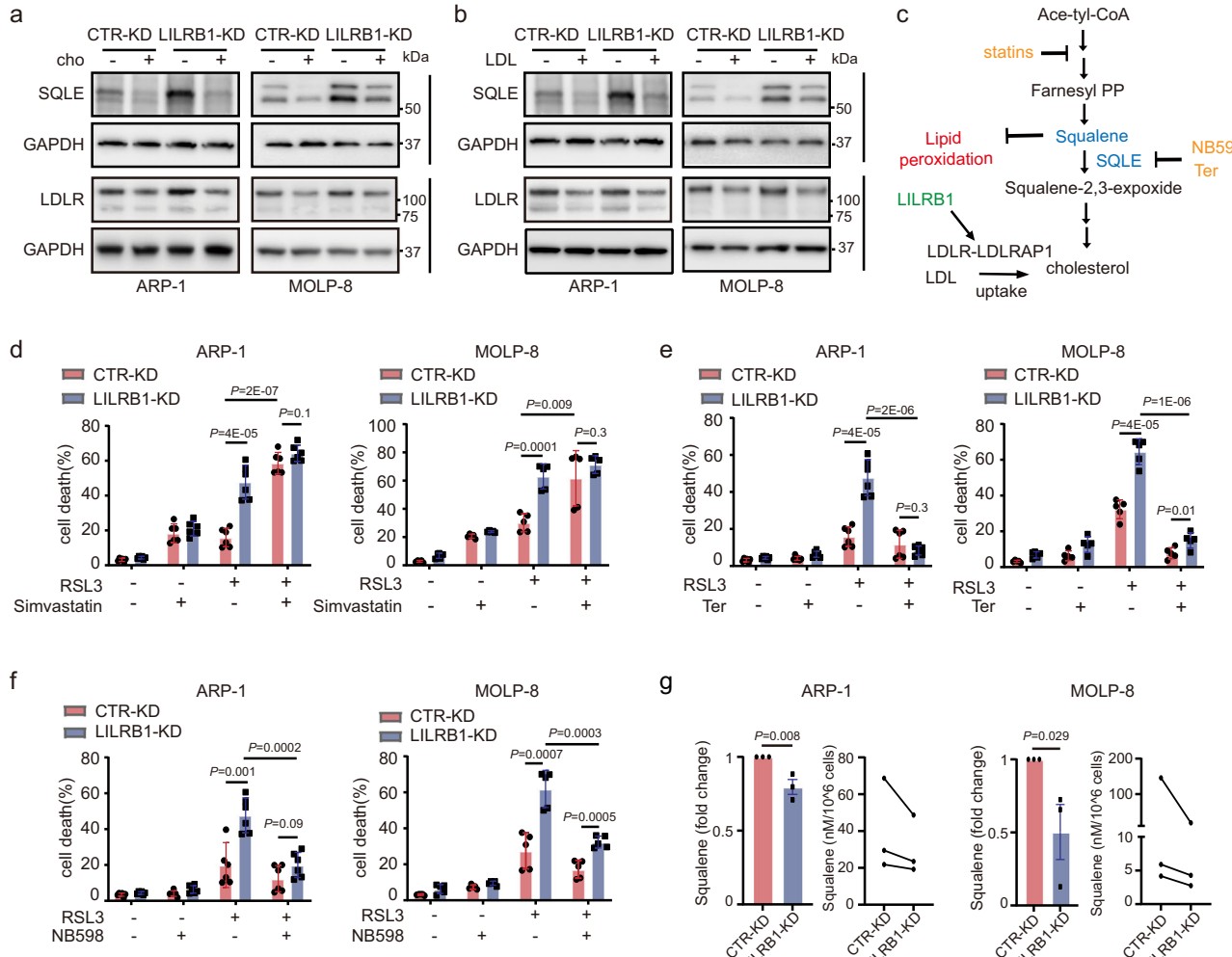

**Fig. 7 | Cholesterol metabolic alteration renders LILRB1-KD MM cells sensitivity to ferroptosis by upregulating SQLE and downregulating squalene.**
**a**, **b** Western blot showing the expression of LDLR, SQLE, GAPDH in CTR-KD or LILRB1-KD MM cells treated with or without cholesterol (**a**) or LDL (**b**). The independent experiments were repeated three times and the representative images are shown. **c** Schematic diagram showing the mechanism underlying LILRB1's role in maintaining cholesterol balance and inhibitors in cholesterol synthesis pathway. **d**–**f** Cell death of CTR-KD or LILRB1-KD MM cells treated with 10 h RSL3 (400 nM) and/or different inhibitors in cholesterol synthesis pathway [simvastatin (10 μM) (**d**), terbinafine (20 μM) (**e**), or NB598 (5 μM) (**f**)]. **g** CTR-KD or LILRB1-KD MM cells were counted and harvested, followed by detection of squalene levels in cells by HPLC-MS. Data are representative of at least three independent experiments and are presented as mean ± SD. (**d**–**f**) -ARP1, $n = 6$; (**d**–**f**)-MOLP-8, $n = 5$; **g**, $n = 3$; $n$, independent experimental repeats; For (**d**–**f**), statistical significance was determined by two-tailed Student $t$-test; For (**g**), statistical significance was determined by one-tailed Student $t$-test. Source data are provided as a Source Data file.

negative regulator and inhibit their immune responses[84]. Thus, inhibiting LILRB1 activity could enhance CD8[+] T cell activation[84]. By delivering LILRB1-targeting PROTACs to the tumor microenvironment, activation of T cells could be enhanced through the degradation of T cell-derived LILRB1. This enhancement would lead to increased IFNγ secretion and enhanced ferroptosis in tumor cells. Concurrently, the degradation of tumor-derived LILRB1 would render tumor cells more susceptible to ferroptosis and effectively impede tumor progression. Thus, LILRB1-targeting PROTACs have the potential to simultaneously enhance the immune system and promote tumor ferroptotic cell death, benefiting MM patients.

## Methods
### Ethical statement
Our research complies with all relevant ethical regulations. The use of primary MM cells is approved by the institutional review board at the Houston Methodist Research Institute. All mouse studies are approved by the Institutional Animal Care and Use Committee of the Houston Methodist Research Institute.

### Primary MM cells
This study was approved by the institutional review board at the Houston Methodist Research Institute. Written informed consent was obtained from all patients. BM aspirates were obtained from patients newly diagnosed with MM. Human lymphocytes were isolated by density gradient centrifugation with Ficoll-Paque PREMIUM (Cytiva, 17544202) and the expression of LILRB1/ lipid peroxidation/cell death of CD138[+] MM cells were detected by flow cytometry (Gating strategies are provided in supplementary Fig. 1a.) No sex or gender analysis was carried out.

### MM cell lines
ARP-1 (CVCL_D523) cells were kindly provided by the Arkansas Cancer Research Center, Little Rock, AR. LP-1 (CVCL_0012), KMS-12-BM (CVCL_1334) and MWD MM.13 cells were kind gifts from Dr. Frederic J. Reu of the Cleveland Clinic63. MOLP-8 (CVCL_2124) cells were a kind gift from Dr. Jinsheng Weng, MD Anderson Cancer Center. ANBL-6 (Cat. # SCC429) was purchased from MilliporeSigma. Other cell lines (MM.1 R (CRL-2975), 8226 (CCL-155), U266 (TIB-196)) were purchased

from the American Type Culture Collection (Rockville, MD). All MM cells were cultured in 1% Pen/Strep, 10% FBS-containing RPMI-1640 medium (MilliporeSigma, R8758). Cells were tested routinely for mycoplasma and were free of contamination at the point of our experiments.

## MM mouse models

NOD-scid IL2rg$^{null}$ (NSG) mice were purchased from The Jackson Laboratory. All mouse studies were approved by the Institutional Animal Care and Use Committee of the Houston Methodist Research Institute. We have complied with all relevant ethical regulations. For MM mouse models in our study, the maximal tumor burden permitted by our ethics committee is the point when they develop signs of hind limb paralysis. In our study, the maximal tumor burden was not exceeded. Mice were housed in a clean facility in 12/12 light/dark cycle, with an ambient temperature of 65–75 °F and 40–60 humidity. Mice were fed with Teklad Irradiated Global Soy Protein-Free Extruded Rodent Diet (2920X, inotivco), with sufficient supply.

For mouse experiments, we repeated our key experiment (the effect of LILRB1 knockdown in vivo) with both female (MOLP-8 models) and male mice (ARP-1 models), and found that knockdown of LILRB1 on MM cells inhibit the progression in vivo in both female and male mice. Therefore, in our system the results are similar between genders. For other in vivo experiments (overexpression of MM1R; treatment of liproxstatin-1; treatment of RSL3; treatment of LDL) we use female mice only.

To determine the role of LILRB1 in MM progression in vivo, 6 to 8 weeks old NSG mice were injected with $2 \times 10^6$ luciferase-expressing CTR-KD or LILRB1-KD APP-1 cells (male) or MOLP-8 cells (female) / CTR-overexpression or LILRB1-overexoression MM.1R (female) cells through the tail vein. From the third week, MM-bearing mice were followed by bioluminescent imaging for the tumor burden if the cells express luciferase. Serum was collected weekly and used for measurement of light chains by ELISA. At the endpoint (hind limb paralysis), mice were sacrificed and the percentage of MM cells (CD138$^+$ or GFP$^+$) in BM was detected by flow cytometry (Gating strategies are provided in supplementary Fig. 9). To detect lipid ROS expression in MM cells in vivo, CTR-KD and LILRB1-KD MM-bearing female mice were sacrificed 7 or 15 days after tumor inoculation, and MM cells in the BM were stained with BODIPY™ 665/676 dye, followed by flow cytometry analysis[88]. To determine whether the inhibition of MM progression induced by the deficiency of LILRB1 in vivo was dependent on ferroptosis, female NSG mice were injected with $2 \times 10^6$ luciferase-expressing CTR-KD or LILRB1-KD ARP-1 cells, followed by administration of vehicle or ferroptosis inhibitor liproxstatin-1 (15 mg/kg, ip) every day and monitor of tumor burden as described before (CTR-KD, $n = 4$; LILRB1-KD, $n = 4$; CTR-KD+liproxstatin-1, $n = 7$; LILRB1-KD +liproxstatin-1, $n = 7$). To assess the impact of LDL on MM cells in mouse model, female NSG mice were injected with $2 \times 10^6$ CTR-KD or LILRB1-KD ARP-1 cells with/without LDL (10 mg/kg, iv) through tail vein, followed by administration of vehicle or LDL (10 mg/kg, iv) biweekly and monitor of tumor burden as described before ($n = 5$). To explore the effect of RSL3 on CTR-KD MM cells and LILRB1-KD MM cells in vivo, female NSG mice were injected with $2 \times 10^6$ CTR-KD or LILRB1-KD ARP-1 cells, followed by administration of vehicle (veh) or RSL3 (20 mg/kg, ip) biweekly and monitor of tumor burden as described before ($n = 5$).

## Knockdown and overexpression of genes

LILRB1-specific shRNA (target sequence: #1- GTCTAA-GATCAACGTACCAAT; #2-GACAGTTCTATGACAGAGTCT; #3- GAACT-CAGGAGGGAATGTAA) was designed and synthesized by Sigma (St. Louis, MO) and inserted into a lentivirus system vector pLVTHM, which was a gift from Didier Trono (Addgene plasmid # 12247). After lentiviral infection, cells were sorted for GFP$^+$ cells followed by western blot

to examine LILRB1 expression. Supplementary Fig.1b, c used MM cells transfected with LILRB1-KD #1–3 viruses, while other experiments related to LILRB1-KD used MM cells transfected with a mixture of LILRB1-KD #1 and #3 viruses, referred to as LILRB1-KD cells. SQLE-specific shRNA (target sequence: #1- GCACCACAGTTTAAAGCAAAT; #2- GCTCAGGCTCTTTATGAATTA) was used for the knockdown of SQLE. LDLR-specific shRNA (target sequence: GGGCGACA-GATGCGAAAGAAA) was used for the knockdown of LDLR. LDLRAP1-specific shRNA (target sequence: GAGAAAGAGAAGAGGGACAAA) was used for the knockdown of LDLRAP1.

LILRB1, LDLR, and LDLRAP1 overexpression were achieved by using lentiviral expression pUltra (carrying GFP selective marker; Addgene plasmid # 24129), a gift from Malcolm Moore, according to the manufacturer's instruction. For overexpression of genes in 293 T cells, we used the pUltra based plasmids. For MM cell lines, we produced lentivirus with pUltra plasmids and infected MM cells with lentivirus. Western blot was used for the detection of LILRB1 expression.

## Gene-expression profiling and analysis of clinical datasets

MM patient microarray datasets and RNAseq data, including array platform information and related clinic factors, were extracted and downloaded from the Oncomine (www.oncomine.org) and MMRF coMMpass IA13 study. For analysis of GEP between MM patients with poor prognosis and MM patients with good prognosis, Zhan's MM 2 dataset was used. MM patients with good prognosis ($n = 54$) were defined as patients who survived more than 4 years from diagnosis, and MM patients with poor prognosis ($n = 54$) were defined as patients who died in less than 2 years. Student $t$-test was used to determine the statistical significance and the difference between the two groups was evaluated and ranked by the value of 2 ^ (average expression value of MM patients with survival < 2 years−average expression value of MM patients with survival ≥ 4 years). For patient survival analysis, overall survival was evaluated in newly diagnosed MM patients based on their LILRB1/LDLR/LDLRAP1 expression status. Patients in the top 25% with highest LILRB1/LDLR/LDLRAP1 expression were categorized as LILRB1/LDLR/LDLRAP1$^{high}$, while the remaining patients were classified as LILRB1/LDLR/LDLRAP1$^{low}$. The number of patients in each group is indicated in the figures. Patient survival curves were plotted using the Kaplan-Meier analysis and significance was measured using the log-rank test.

## RNAseq analysis

RNAseq of LILRB1-KD and CTR-KD ARP-1 cells was performed by Cancer Genomics Center at UTHealth. The DEseq2 method was used for differential expression analysis. IPA analysis and GSEA of CTR-KD and LILRB1-KD MM cell RNAseq data were performed for assessment of dysregulated signaling and metabolic pathways in LILRB1-KD MM cells versus CTR-KD MM cells. GSEA was run for each cell subset in pre-ranked list mode with 1000 permutations (nominal $p$-value cutoff < 0.01). RNA-seq data have been deposited into the RNAseq of Gene Expression Omnibus database with accession number GSE226821.

## Mass spectrum analysis

Mass spectrum analysis of anti-IgG ($n = 1$) and anti-LILRB1 ($n = 1$) co-IP pull-down products of ARP-1 cells were performed and analyzed by the Clinical and Translational Proteomics Service Center at the Institute of Molecular Medicine, University of Texas. GO enrichment analysis was used for analyzing the biological process in which potential LILRB1-interacting proteins may be involved.

MS instruments used for the experiment: Orbitrap Fusion™ Tribrid™ mass spectrometer (Thermo Scientific™) interfaced with a Dionex UltiMate 3000 Binary RSLCnano System.

Sample processing protocol: The peptides were analyzed using data-dependent acquisition method, Orbitrap Fusion was operated

with measurement of FTMS1 at resolutions 120,000 FWHM, scan range 350-1500 m/z, AGC target 2E5, and maximum injection time of 50 ms; During a maximum 3 s cycle time, the ITMS2 spectra were collected at rapid scan rate mode, with HCD NCE 34, 1.6 m/z isolation window, AGC target 1E4, maximum injection time of 35 ms, and dynamic exclusion was employed for 35 s.

Data processing protocol: The raw data files were processed using Thermo Scientific™ Proteome Discoverer™ software version 1.4, spectra were searched against the database using Sequest HT search engine. The spectra were also searched against decoy database using a target false discovery rate (FDR) of 1% for strict and 5% for relaxed conditions. For the trypsin, up to two missed cleavages were allowed. MS tolerance was set 10 ppm; MS/MS tolerance 0.6 Da. Carbamidomethylation on cysteine residues was used as fixed modification; oxidation of methione as well as phosphorylation of serine, threonine and tyrosine was set as variable modifications.

## Antibodies

**For western blot.** Anti-FLAG (66008-4-Ig, Lot 10027647;20543-1-AP, Lot 00106091), anti-LDLR (10785-1-AP, Lot 00118477), anti-LDLRAP1 (66932-2-Ig, Lot 10008603), anti-APOB (20578-I-AP, Lot 00117266), anti-APOE (66830-I-Ig, Lot 10008911), anti-HMGCR (13533-1-AP, Lot 00120917) and anti-His (10001-0-AP, Lot 00101471; 66005-1-Ig, Lot 10020245) antibodies were purchased from Proteintech. Anti-LILRB1 antibody was purchased from cell signal technology (LILRB1/CD85j (D4L8L) Rabbit mAb, 78144) and abcam (Anti-LILRB1 antibody [EPR22861-6], ab238145, Lot GR3416682-2). Anti-SOLE (sc-271651, Lot # A0421) and anti-GAPDH (sc-32233), Lot # I0319 antibodies were purchased from Santa Cruz Biotechnology. Anti-mouse (NA9311ML, Lot 17170538) and anti-rabbit (NA9341ML, Lot 17271476) secondary antibodies were purchased from Cytiva.

**For immunoprecipitation.** Anti-FLAG antibody was purchased from Sigma-Aldrich (F1804, clone M2). Anti-LILRB1 antibody (ab238145, Lot GR3416682-2) and recombinant rabbit IgG isotype control were purchased from Abcam. Anti-His (66005-1-Ig, Lot 10020245) and anti-LDLR (10785-1-AP, Lot 00118477) were purchased from Proteintech. Normal rabbit IgG was purchased from CST.

**For immunofluorescence (IF).** Anti-LILRB1 antibody (rabbit, ab238145, Lot GR3416682-2) was purchased from Abcam. Anti-LDLR antibody was purchased from R&D (goat, AF2148, Lot VBC0221081); anti-LDLRAP1 antibody was purchased from Santa Cruz Biotechnology (mouse, sc-514263, Lot #B2422). Donkey anti-goat secondary antibody (AF647; Invitrogen, A32849, Lot: WL333743), donkey anti-mouse secondary antibody (AF555; Invitrogen, A32773, Lot: XC344355), and donkey anti-rabbit secondary antibody (AF488; Invitrogen, A-21206, Lot:2376850) were purchased from Invitrogen.

**For flow cytometry.** APC Annexin V, biolegend, 640941, Lot: B386061; APC anti-human CD85j (ILT2) Antibody, biolegend, 333720, clone GHI/75, Lot: B316130; APC Mouse IgG2b, κ Isotype Ctrl Antibody, biolegend, 400322, clone MPC-11, Lot: B317316; PE anti-human CD85j (ILT2) Antibody, biolegend, 333708, clone GHI/75, Lot: B274730; PE Mouse IgG2b, κ Isotype Ctrl Antibody, biolegend, 400314, clone: MPC-11; APC anti-human CD138 (Syndecan-1) Antibody, biolegend, 352308, clone: DL-101, Lot: B337398; PE anti-human CD138 (Syndecan-1) Antibody, biolegend, 352306, clone: DL-101, Lot: B259247; PE anti-Apo E Antibody, biolegend, 803405, clone: E6D7, Lot: B278267.

## MTS assay

To examine the relative changes of cell viability among different groups, LILRB1-KD or CTR-KD MM cells ($2 \times 10^5$ /ml) were planted into a 96 well-plate and cultured for 72 h. An MTS assay (Promega, Madison,

WI) was conducted based on the manufacturer's protocol. Results were normalized to CTR-KD MM cells.

## Detection of cell death

MM cells were harvested and washed with PBS buffer, followed by staining in PI solution (1:100 diluted in PBS buffer) for 15 min. PI positive cells were detected and analyzed to show the percentage of cell death by flow cytometry (BD FACS Symphony A3, BD Biosciences).

RSL3, FIN56, erastin, and AA were used to induce ferroptotic cell death. Simvastatin, NB 598, Terbinafine HCl, cholesterol, and LDL were used to test their effects on ferroptotic cell death. Ferroptosis inhibitors liproxstatin-1, deferoxamine mesylate and ferrostatin-1 were used for the inhibition of ferroptosis. Gating strategies are provided in supplementary Fig. 9.

## Analysis of squalene by HPLC-MS

Squalene was extracted from cell samples using 700 mL acetone:methanol at 1:1 (v:v) ratio. Ten mL internal standard squalene-d6 (400 ng/mL) was added to each cell sample. Samples were vortexed for 10 min, centrifuged at 2,500 g for 20 min at 4 °C, and supernatants were transferred to clean tubes, followed by evaporation to dryness under nitrogen. Samples were reconstituted in 100 mL methanol, and 10 mL of the solution was injected into a Thermo Vanquish ultra-high pressure liquid chromatography (UHPLC) system containing a Fortis SpeedCore PFP column (2.1 × 50 mm, 2.6 mm). Mobile phase A (MPA) was water and mobile phase B (MPB) was methanol. The flow rate was 300 mL/min (at 35 °C), and the gradient conditions were: 0–3 min, 65% to 95% MPB, 3–4 min, 95% MPB, 4.1–7 min, 65% MPB. The total run time was 7 min. Data were acquired using a Thermo Orbitrap Fusion Lumos Tribrid mass spectrometer under APCI positive ionization at a resolution of 120,000 in selected ion monitoring (SIM) mode. Squalene and squalene-d6 were monitored at 411.3985 and 417.4362 m/z respectively. Raw files were imported to Thermo Trace Finder software for final analysis. The concentrations of squalene were quantitated against an external calibration curve.

## Lipid peroxidation detection

Experiments were performed according to the manufacturer's protocols:

**BODIPY™ 665/676 assay assay and BODIPY™ 581/591 C11 assay.** BODIPY™ 665/676 dye (Lipid Peroxidation Sensor) and BODIPY™ 581/591 C11 dye (Lipid Peroxidation Sensor) were purchased from Thermo Fisher Scientific. Briefly, cells were incubated in a humidified chamber at 37 °C with 5% CO2 for 30 min with BODIPY™ 665/676 or BODIPY™ 581/591 C11 in cell culture medium. After incubation, cells were washed and examined by flow cytometry within 2 h of staining. As BODIPY™ 665/676 or BODIPY™ 581/591 C11 exhibits a change in fluorescence after interaction with peroxyl radicals, the lipid ROS levels measured by BODIPY™ 665/676 were presented by mean fluorescence (PE-CF594) divided by mean fluorescence (APC) and the lipid ROS levels measured by BODIPY™ 581/591 C11 were presented by mean fluorescence (FITC) divided by mean fluorescence (PE). Gating strategies are provided in supplementary Fig. 9. The summary results were normalized to the untreated control cells.

**MDA assay.** Lipid peroxidation (MDA) assay kit (ab233471) was purchased from Abcam. Cells were lysed by homogenization. Lysates were centrifuged to remove cell debris (5 min at 10,000 × g) and the supernatant was used for deproteinization using a deproteinizing sample preparation Kit (ab204708). After deproteinization, the MDA color reagent was added to samples and incubated for 30 min at room temperature. The reaction solution was added and incubated for 60 min at room temperature, followed by analysis with microplate

reader at 695 nm. The results were normalized to untreated control cells.

## Immunofluorescent assay

Co-localization of LILRB1, LDLR, and LDLRAP1 in MM cells was examined by confocal microscope after staining fixed MM cells with primary antibodies (anti-LILRB1 antibody, Abcam, Rabbit; anti-LDLR antibody, R&D, Goat; anti-LDLRAP1, Santa Cruz, Mouse), followed by staining with a secondary antibody. The uptake of LDL was examined by confocal microscope after incubating MM cells with pHrodo™ Red-LDL (Thermo fisher scientific, L34356) for 2 h. Co-localization of LILRB1 and LDL was examined by confocal microscope after staining live cells with anti-LILRB1 antibody and pHrodo™ Red-LDL(L34356).

## Western blotting analysis

Cell lysates and immunoblotting were performed as previously described[89]. Cells were harvested and lysed with Cell Lysis Buffer (Cell Signaling Technology) supplemented with protease/phosphatase inhibitor cocktail (Cell Signaling Technology). The Bradford Protein Assay (Bio-Rad) was used to detect the protein concentration of the samples. Protein samples were then subjected to NuPAGE 4–12% Bis-Tris Protein Gels (Invitrogen), transferred to a nitrocellulose membrane (Bio-Rad), and immunoblotted with primary antibodies overnight. Secondary antibodies conjugated to horseradish peroxidase were used for detection, followed by enhanced chemiluminescence (Pierce Biotechnology).

## Soft agar colony-formation assay

Agar solution at 2.5% was prepared by dissolving 0.5 g low melting point agar (sigma, A9045) in 50 mL of deionized water followed by autoclaving at 121 °C for 20 min. After that, 8 mL cell culture medium (37 °C) was added into 2 mL 2.5% agar solution (50 °C) and well-mixed solution was dispensed into a 6-well plate. The plate was placed at 4°C to solidify quickly and then rewarmed at 37 °C in a cell incubator. To make the up layer, 9.4 mL complete cell culture medium (37 °C) was added to 1.2 mL 2.5% agar solution (50 °C) and mixed well. The mixture containing 1.5 ml up-layer solution with 100 ml resuspended cells (2500/well) was added onto the solidified bottom layer of agar, allowed to solidify for 30 min at room temperature, and cultured in a cell incubator for 2-3 weeks before counting and imaging.

## LDL uptake assay by labeled LDL and flow cytometry

Experiments were performed according to the manufacturer's protocol. Briefly, MM cells were cultured in FBS-free cell culture medium for 24 h. After that, pHrodo™ Red-LDL (Thermo Fisher Scientific, L34356) was added to FBS-free cell culture medium (1:500 dilution) and cultured for 2 h. The amount of LDL uptake in MM cells was measured by flow cytometry or fluorescence microscope without wash steps.

## Detection of extracellular (uptake of LDL/cholesterol) and intracellular LDL/cholesterol

Multiple myeloma (MM) cells were seeded in lipoprotein-free medium, and LDL was subsequently added. A non-uptake control group (with LDL in lipoprotein-free medium and without MM cells) was included for comparison. After a 2 day incubation period, we collected both the culture medium and the MM cells, measuring LDL/cholesterol concentrations in both the extracellular medium and intracellular MM cells by total cholesterol assay kit (STA-390, CELL BIOLABS, INC.). Experiments were performed according to the manufacturer's protocol. The calculation for LDL/cholesterol uptake by MM cells is as follows: subtracting the concentration in the medium with MM cells from the concentration in the medium without MM cells.

## Detection of CoQ10 by ELISA kit

Human Coenzyme Q10 (CoQ10) ELISA Kit (Colorimetric) (NBP3-21147, Novus) was used to detect the levels of CoQ10 in CTR-KD MM cells and LILRB1-KD MM cells. Experiments were performed according to the manufacturer's protocol.

## Analysis of Squalene and CoQ10 by UHPLC-HRMS (isotope tracing analysis)

CTR-KD ARP-1 cells and LILRB1-KD ARP-1 cells were cultured in culture medium with $^{13}C_6$-glucose for 24 h and 20-30 *10 ^ 6 cells were harvested for each sample to send for isotope tracing analysis, which is performed and analyzed by the Metabolomics Core Facility at the University of Texas MD Anderson Cancer Center. To determine the incorporation of glucose carbon ($^{13}C_6$-glucose) into intracellular Squalene/CoQ10, Cell extracts were prepared and analyzed by UHPLC/high-resolution mass spectrometry (HRMS).

Squalene was extracted from cell pellets using 1/1 (v/v/) acetone/methanol. Samples were vortex for 10 min and then centrifuged at 17,000 g for 10 min at 4 °C, and supernatants were transferred to clean tubes, followed by evaporation to dryness under nitrogen. Samples were reconstituted in methanol, then 10 μl was injected into a Thermo Scientific Vanquish UHPLC system containing a Fortis FPF 50 × 2.1 mm 2.6 μm column. Mobile phase A was water and mobile phase B was methanol. The flow rate was 350 μL/min (at 35 °C) and the gradient conditions are as follows: 0 min 65%B, 3 min, 95% of B, 3-6 min, 95%, 6.1–8 min 65% of B. The total run time is 8 min. Data were acquired using a Thermo Orbitrap Lumos Tribrid Mass Spectrometer under APCI positive mode. Then the raw files were imported to Skyline-Daily software for final analysis.

CoQ10 was extracted from cell pellets using 1/1 (v/v/) acetone/methanol. Samples were vortexed for 10 min and then centrifuged at 17,000 g for 10 min at 4 °C, and supernatants were transferred to clean tubes, followed by evaporation to dryness under nitrogen. Samples were reconstituted in methanol, then 10 μl was injected into a Thermo Scientific Vanquish UHPLC system containing a Phenomenex Luna C5 50 × 4.6 mm 5 μm column. Mobile phase was 5 mM ammonium formate in methanol. 100% of mobile phase was delivered at 400 μL/min (at 30 °C) isocratically. The total run time is 10 min. Data were acquired using a Thermo Orbitrap Lumos Tribrid Mass Spectrometer under ESI positive mode. Then the raw files were imported to Skyline-Daily software for final analysis.

## Statistics & Reproducibility

Sample size was determined to be adequate based on the magnitude and consistency of measurable differences between groups. For in vivo mouse experiments, mice were randomly grouped prior to be treated. The investigators were not blinded to allocation during experiments and outcome assessment. The statistical analysis was performed using GraphPad Prism (version 9.0.0) and Excel 2019. Statistical tests used in the different experiments are indicated in the respective figure legends.

## Reporting summary

Further information on research design is available in the Nature Portfolio Reporting Summary linked to this article.

## Data availability

All raw sequencing data generated in this study have been deposited in the NCBI Gene Expression Omnibus (GEO) under accession number GSE226821. The mass spectrometry proteomics data (Mass spectrum analysis to identify the interacting protein of LILRB1 in multiple myeloma cell line ARP-1) have been deposited to the ProteomeXchange Consortium via the PRIDE[90] partner repository with the dataset identifier PXD045817. Other previous published MM patient datasets used in the study can get access with the following accession number and link: GSE2658; GSE4452; GSE5900; GSE19784; phs000748. Source data are provided with this paper. All the other data supporting the findings of this study are available with in the article, supplementary information, source files. Source data are provided with this paper.

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

## Acknowledgements

This work was supported by Startup Support from Houston Methodist Research Institute, Houston Methodist Hospital, by Cancer Prevention & Research Institute of Texas Recruitment of Established Investigator Award (RR180044) and High-Impact/High-Risk Research Award (RP210868), and by NCI R01s CA211073 and CA214811 to Q.Y. Q.Y. and his research group were also supported by NCI R01s (CA200539, CA239255, CA282099, CA278787, CA2855209, and CA285203 to Q.Y.). The authors thank Dr. Philip L Lorenzi and Dr. Bo Wei and Metabolomics Core Facility at the University of Texas MD Anderson Cancer Center for their services and support on the analysis of squalene quantification, as well as isotope tracing of squalene and CoQ10. This work is supported in part by the Clinical and Translational Proteomics Service Center at the University of Texas Health Science Center.

## Author contributions

Q.Y., M.X. and Q.W. initiated the study. Q.Y., M.X. and Q.W. designed the research and wrote the paper. M.X., Q.W., L.X., L.Z., W.X., C.Z., J.Q, and Y.L. performed experiments; Y.Z., R.O. and F.Z. provided experimental support; Y.Z. provided patient samples; and L.Z., P.S., L.X., W.X., C.Z., R.O., F.Z., J.Q. and Y.L. provided critical suggestions. R.O., F.Z., Q.W., and M.X. planned and performed the bioinformatics analysis of the clinical microarray datasets. Q.W., M.X., and L.Y. planned and performed the analysis of RNAseq data and Mass spectrum data.

## Competing interests

The authors declare no competing interests.
