## [Peer Review File · Nature Communications]

Leukocyte immunoglobulin-like receptor B1 (LILRB1) protects human multiple myeloma cells from ferroptosis by maintaining cholesterol homeostasisREVIEWER COMMENTS

Reviewer #1 (Remarks to the Author):

This manuscript explores the proliferation of multiple myeloma from the perspective of cholesterol metabolism, which has certain clinical significance and novelty. However, the following concern need to be addressed:

Major revision:

1. The author should further detect the concentration of intracellular and extracellular LDL in MM cell lines with overexpressing and knocking out LILRB1, thereby proving that LDL is indeed uptake through LILRB1.

2. It is recommended that the author need to label LDL into tumor cell lines through metabolic flow experiments to detect whether the final metabolites are related to ferroptosis or proliferation.

3. If patient samples can be supplemented to detect whether there is a change in the expression of iron death related proteins in MM cells in patients with high LDL, this can be further verified the clinical relevance.

4. It is recommended that the author perform in vivo study to observe the tumor burden of the MM mouse model through LDL treatment and ferroptosis inhibitors.

Reviewer #2 (Remarks to the Author):

In this manuscript Xian et al. identified LILRB1 as one the 20 genes that were significantly overexpressed in multiple myeloma (MM) patients with <2-year survival vs. those surviving more than 4 years. Collectively, they provide detailed mechanistic insight into the regulation of MM cell survival by LILRB1. Specifically, they report that genetic knock-down of LILRB1 in MM cell lines significantly delays tumor growth and survival in vivo. Their proteomics analysis reveals that LILRB1 interacts with LDLRAP1 and LDLR, which facilitates LDL uptake and protection from ferroptotic cell death. This is relatively a well-designed study which identifies a novel role for LILRB1 in the pathogenesis of MM. However, I have a number of reservations and comments, which should be addressed before further consideration. My major suggestion is that the authors should use primary cells to confirm some of the key findings with the MM cell lines.

Major comments:

1- Figure 1J and Lines 109-12: The expression of LILRB1 is very heterogeneous on primary MM cells- the authors need to comment on this. What gating was used here- ideally the gating strategy should be shown, perhaps as supplemental data. Would sorting positive and negative cells show differences between the two subpopulations, as the show with LILRB1 negative and positive MM cell lines? Not all samples express 'LILRB1' highly, as the text suggests.

2- Figures: the n numbers and number of independent experiments and mice/group should be clearly indicated throughout. As it stands, most figures are n=1 and need to be repeated.

3- Figure 7: WB images in A and B, do not convincingly indicate that SQLE is higher in LILRB1-KD MM cells, in particular in MOLP-8. The same applies to the WB image shown in Sup Fig. 7B.

Overall, I had a hard time interpreting the results shown in Fig 7, which may be due to the lack of my expertise in the pathway shown.

4- To confirm the in vitro observations, it would be helpful if MM-bearing mice were treated with RSL3 or similar.

Minor comments:

1- Discussion: LILRB1 is an inhibitory receptor with multiple ITIMs, and previous studies indicated that LILRB1 inhibits B cell signaling/functions (reviewed in doi: 10.1172/jci.insight.151553).

However, here it is argued that LILRB1 protects the MM cells and provide a survival advantage, so the authors should discuss the discrepancies between these observations.

2- Line 50: provide references.

3- Line 53: include citations showing negative regulation of macrophages by LILRB1 and put more

emphasis on immune-evasion.

4- Lines 83-84: The grammar for 'longer-survived' and 'poorly survived' should be corrected throughout the text and in the figures.

5- Lines 89-90: They should justify why LILRB1 was chosen out of all 20 upregulated genes and link to previous literature and its characteristics, eg, cell surface expression.

6- Figure 1B: the expression of LILRB1 and other genes seems quite variable among donors, and is up- or down-regulated in poor vs long survivals- the authors should comment on this.

7- Figure 1: Graphs should be aligned.

8- Figure 1G and H: How did they define LILRB1 high vs low? The bioinformatics analyses should be described in the M&M section.

9- M&M: Indicate the clone of antibodies used throughout.

10- M&M: indicate the sex of mice used.

11- M&M: provide a summary of the patient characteristics and clinical details, where possible.

12- Figure 3: The size of text in the figure should be increased.

13- Figure 4 and lines 192-193: It is not clear what cell line was used for the co-IP assay in A. Was it LILRB1-transfected 293T cells? If so, this is not the best cell line to use and ideally this should be repeated with MM cell lines.

14- Line 219: replace 'discover' with an alternative word, such as 'investigate'.

15- Sup Fig 8: a descriptive text should be included to summarize the schematic diagram.

16- Line 335: correct grammar- 'haven't' to 'have not'.

17- Supp Table 1 and 2: A number of other LILR family members, including LILRB4, appear in the list of genes indicated in S. Table 1. Similarly, LILRB4 comes up in the proteomic analysis. The authors should comment on this, if significant. Do LILRB1 and LILRB2 form heterodimers?

Reviewer #3 (Remarks to the Author):

In this study, Xian et al identified a novel factor, LILRB1, that is involved in the prognosis of multiple myeloma and demonstrated that LILRB1 expression is important for tumor growth in vivo. Through transcriptome analysis, the authors found differences in fatty acid metabolism and oxidative stress response in LILRB1 KD cells, leading them to identify LILRB1 as a regulator of ferroptosis. The identification of a novel player in ferroptosis is always important and interesting; however, the proposed mechanisms are not entirely supported by the data. There are also several important issues that must be addressed.

Major concerns

1. The study, which found that accumulated squalene can inhibit ferroptosis, was limited to lymphomas lacking SQLE and may not be applicable in the context of multiple myeloma. In normal cancer cells, the mevalonate pathway operates rapidly, and as a result, squalene is generally undetectable in the cell. What is the level of SQLE expression in the various MM cells depicted in Fig. 1i? Do the two cells the authors used, ARP-1 and MOLP-8, have high levels of squalene with low SQLE levels of activity? It is necessary to compare the amount of squalene in MM cells and present absolute amounts, particularly in Fig. 7g. In addition, the authors must demonstrate the levels of cholesterol in cells.

2. While the authors used standard ferroptosis inducers like RSL3, FIN56, and erastin, they did not confirm whether ferroptosis-specific inhibitors like ferrostatin-1 or liproxstain-1 can completely rescue cell death and lipid peroxidation. Please also specify how long cells were treated with ferroptosis inducers for cell death and lipid peroxidation analysis.

3. Regarding arachidonic acid (AA) treatment for ferroptosis, AA does not normally induce ferroptosis on its own, but it sensitizes cells to ferroptosis when combined with other ferroptosis inducers. Furthermore, when compared to other ferroptosis studies and considering the physiological concentration of AA, 75 nM AA is too low to induce or sensitize cells to ferroptosis. Ferroptosis inhibitors should be used to confirm AA experiments.

4. If, as the authors claim, LDLR uptake of LDL and cholesterol is the primary mechanism, these experiments should be validated in lipoprotein-deficient serum. Does a lack of lipoproteins cause

ferroptosis by upregulating SQLE?

5. Indeed, HMGCR is more well-known as the rate-limiting enzyme in the mevalonate pathway than SQLE, but its involvement in this pathway has not been investigated. The authors must provide HMGCR levels upon LILRB1 KD. Furthermore, SQLE upregulation may accumulate CoQ10, which may contribute to ferroptosis suppression. It is difficult to understand how changes in SQLE expression affect each intermediate in the mevalonate pathway without metabolic tracing analysis.

6. Because so many drugs can inhibit ferroptosis via non-specific RTA activity, it is difficult to determine whether Ter and NB598's ferroptosis inhibitory effects are due to SQLE inhibition or RTA function. Indeed, the fact that both drugs inhibit ferroptosis in control cells suggests that it is most likely SQLE-independent, or that basal levels of squalene play a role in ferroptosis independent of LILRB1. At the very least, please include the concentrations of the two drugs you used.

7. In Fig. 2, the mechanism by which LILRB1 affects tumor growth in vivo is completely unknown in Fig. 2. Is this phenotype caused by ferroptosis or squalene metabolism? The authors have to supply molecular details as well as rescue experiments using ferroptosis inhibitors.

8. The authors must be truthful when citing references. Many references are either irrelevant or have been replaced by review articles. References 10, 39, 40, and 41, in particular, should be replaced with original papers. The references 11 and 12 for AA are not experimental papers.

Minor concerns

9. In Fig. 3g,m,p,r,t, please display the histogram data for untreated cells.

10. In Fig. 3t, Given the difference in area between the two groups, the number of cells measured in each group is likely to be different.

11. In the legend of Fig. 5, LDL concentration might be "uM".

12. In Fig. 6, 7, the WB band patterns of SQLE blots are inconsistent. The upper band accumulated specifically upon KD of the gene in some blots, but the opposite result also occurred.

13. Do not truncate the scale bar on the y-axis, as this magnifies the differences between groups.

Reviewer #4 (Remarks to the Author):

The current study by Miao et al found that LILRB1 associated with poor prognosis in multiple myeloma. Knockdown of LILRB1 increased expression of cholesterol biosynthetic genes and ferroptosis. They found that LILRB1 may interact with LDLR and LDLRAP1 to promote LDL uptake. This study is interesting but further experiments are required to address some major concerns.

1. LILRB1 interacts with LDLR and LDLRAP1 (Fig 4). The authors should map the precise binding sites in these proteins. Does LILRB1 bind to LDLR and LDLRAP1 simultaneously? Are their interactions regulated by LDL?

2. It is important the LILRB1 promotes LDL uptake. Is this process dependent on LDLR and LDLRAP1? Since Dab2 is another adaptor protein for LDLR internalization, does Dab2 plays a role in LILRB1-mediated LDL uptake? Can LILRB1 directly mediate LDL uptake? Did the LILRB1 knock-down alter LDLR expression?

3. The connection between MM poor prognosis, LILRB1 high expression and ferroptosis is weak.

4. If the hypothesis of this manuscript is correct. The connection between MM and LDLR, PCSK9 and other cholesterol-metabolic genes mutations should be observed. Can the authors provide this

information?

5. The title of this study should be revised since no clinic applicable targeting approach was used in this study.

Responses to Reviewers' Comments

We would like to thank the reviewers for careful and thorough reading of this manuscript and for the thoughtful comments and constructive suggestions, which have helped to improve the quality of this manuscript. The corresponding changes and refinements made in the revised paper are summarized in our response below. Please note that the specified changes/modifications in the revised version of the manuscript are highlighted by underlined red texts.

Reviewer Comments:

Reviewer #1 (Remarks to the Author):

1. The author should further detect the concentration of intracellular and extracellular LDL in MM cell lines with overexpressing and knocking out LILRB1, thereby proving that LDL is indeed uptake through LILRB1.

Thanks for the suggestions. We detected the concentration of intracellular and extracellular LDL/cholesterol in LILRB1-overexpressing and LILRB1-KD MM cells. Our results demonstrated that LILRB1-KD cells uptake less LDL/cholesterol than CTR-KD cells (**Fig. R1-1 a**, below, the same thereafter), as we detected more cholesterol in the supernatant of LILRB1-KD cells. The intracellular cholesterol level did not show significant difference between CTR-KD cells and LILRB1-KD cells (**Fig. R1-1 b**). These data demonstrate that LILRB1-KD could inhibit the uptake of LDL/cholesterol in MM cells and suggest that LILRB1-KD cells may activate the synthesis pathway to keep the balance of cholesterol, which is consistent with our findings that the cholesterol synthesis enzyme SQLE was upregulated in LILRB1-KD cells. On the other hand, LILRB1-overexpressing MM cells uptake more LDL/cholesterol (**Fig. R1-1 c**), while there was no significant difference in the intracellular cholesterol between LILRB1-overexpressing and CTR MM cells (**Fig. R1-1 d**). Taken together, these data indicate that LILRB1 could help LDL/cholesterol uptake.

These data are included in the revised version of Fig. 4.

Fig. R1-1 Uptake of LDL/cholesterol and intracellular cholesterol in LILRB1-KD and LILRB1-OE MM cells. (**a, c**) The uptake of LDL/cholesterol was detected by the changes of extracellular LDL/cholesterol concentrations in the supernatant of CTR-KD and LILRB1-KD ARP-1 cells (**a**), as well as CTR and LILRB1-OE MM.1R cells (**c**), by fluorescence microplate reader. (**b, d**) Intracellular cholesterol concentrations of CTR-KD and LILRB1-KD ARP-1

cells **(b)**, as well as CTR and LILRB1-OE MM.1R cells **(d)** were detected. Data are summary of three independent experiments. Statistical significance was determined by Student t test. **p< 0.01; ***p< 0.001; n.s., not significant

2. It is recommended that the authors need to label LDL into tumor cell lines through metabolic flow experiments to detect whether the final metabolites are related to ferroptosis or proliferation.

Thanks for the suggestion. With ¹³C-glucose as a tracer, we traced the squalene and CoQ10 in CTR-KD and LILRB1-KD ARP-1 cells. Our results showed that labeled squalene and CoQ10 were downregulated in LILRB1-KD compared to CTR-KD ARP-1 cells **(Fig. R1-2)**, supporting our findings that squalene and CoQ10 were downregulated by the upregulation of SQLE in LILRB1-KD MM cells, leading to more ferroptosis.

These data are included in the revised version of Supplementary Fig. 7.

Fig. R1-2 Isotope tracing of squalene and CoQ10. **(a)** Schema of analysis of [¹³C]glucose uptake to trace incorporation of ¹³C into newly synthesized squalene and CoQ10. **(b-d)** CTR-KD and LILRB1-KD ARP-1 cells were cultured in [¹³C]glucose-containing medium for 24 h and harvested for isotope tracing analysis. **(b)** The labeled fractions of Squalene (M+26) were shown. n=3, biological repeats. **(c)** The labeled fractions of Squalene isotopomers (M+22, ..., M+28) were shown. **(d)** The labeled fractions of CoQ10 (M+16) were. n=2, biological repeats.

3. If patient samples can be supplemented to detect whether there is a change in the expression of iron death related proteins in MM cells in patients with high LDL, this can be further verified the clinical relevance.

Thanks for the suggestions. We examined the datasets of the Multiple Myeloma Research Foundation (MMRF) and identified 44 MM patients who experienced hypercholesterolemia, a disorder known for an excess of LDL in blood¹.

Comparing the gene expression profiles of these 44 MM patients with hypercholesterolemia to 44 MM patients with normal LDL levels, interestingly, MM patients with hypercholesterolemia exhibited a lower expression of pathways associated with ferroptosis activation compared to those with normal cholesterol levels **(Fig. R1-3)**. This finding aligns with our in vitro data and suggests that cholesterol can protect MM cells from ferroptosis.

These data are included in the revised version of Fig. 6.

Fig. R1-3 Gene set enrichment analysis of positive regulation of ferroptosis process genes between MM patients with hypercholesterolemia and normal MM patients in MMRF dataset.

4. It is recommended that the author perform in vivo study to observe the tumor burden of the MM mouse model through LDL treatment.

Thanks for the suggestion. To assess the impact of LDL on MM cells in mouse model, we established four groups:

- a. CTR-KD group (n=5): NSG mice injected with CTR-KD ARP-1 cells, treated with PBS biweekly.
- b. CTR-KD+LDL group (n=5): NSG mice injected with CTR-KD ARP-1 cells and LDL, treated with LDL biweekly.
- c. LILRB1-KD group (n=5): NSG mice injected with LILRB1-KD ARP-1 cells, treated with PBS biweekly.
- d. LILRB1-KD+LDL group (n=5): NSG mice injected with LILRB1-KD ARP-1 cells and LDL, treated with LDL biweekly.

Both the bioluminescent images and summarized results revealed that LDL treatment increased tumor burden in MM-bearing mice (**Fig. R1-4 a,b**). While LILRB1-KD MM cells showed delayed progression in vivo, LDL treatment significantly promoted MM progression in both LILRB1-KD and CTR-KD MM-bearing mice. Tumor burden measured by kappa light chain also supported these findings (**Fig. R1-4 c**). These in vivo results align with our in vitro data, indicating that LDL protects both CTR-KD and LILRB1-KD MM cells from ferroptotic cell death.

These data are included in the revised version of Supplementary Fig. 7.

Fig. R1-4 Effect of LDL on MM cells in vivo. NSG mice were injected with 2×10^6 CTR-KD or LILRB1-KD ARP-1 cells with or without LDL (10 mg/kg, iv) through tail vein, followed by administration of vehicle (veh) or LDL (10 mg/kg, iv) biweekly and monitoring of tumor burden (n=5). Representative bioluminescent imaging for tumor burden (**a**) and summarized quantification of bioluminescent imaging (**b**) are shown. (**c**) Tumor burden was measured as serum concentration of κ light chain. Statistical significance was determined by Student t test. * $p < 0.05$; ** $p < 0.01$; *** $p < 0.001$.

Reviewer #2 (Remarks to the Author):

Major comments:

1- Figure 1J and Lines 109-12: The expression of LILRB1 is very heterogeneous on primary MM cells- the authors need to comment on this. What gating was used here- ideally the gating strategy should be shown, perhaps as supplemental data. Would sorting positive and negative cells show differences between the two subpopulations, as the show with LILRB1 negative and positive MM cell lines? Not all samples express 'LILRB1' highly, as the text suggests.

Thanks for the valuable comments.

The expression of LILRB1 is heterogeneous across patient samples, with a common occurrence in primary patient samples. While most primary MM patient samples exhibited expression of LILRB1 (as shown in Fig. 1J, 21 out of 24 samples), the expression levels of LILRB1 varied within each sample. In some patient samples, a portion of cells expressed LILRB1 while another portion did not. Alternatively, a heterogeneous expression pattern may be observed with some cells showing high expression and others showing low expression. **We include this in our revised manuscript (line 126-131).**

The gating strategy is shown below: we first gated FSC-SSC to exclude cell debris, and then CD138 positive cells were gated as MM cells to show the expression of LILRB1 (**Fig. R2-1-1**). We included the gating strategy in **Supplementary Fig. 1**.

gating strategy

Fig. R2-1-1 Gating strategy of Fig. 1j (the detection of the expression of LILRB1 in MM cells in bone marrow leukocytes of MM patients)

We acknowledged the necessity of involving more primary MM patient cells in our study. However, we faced challenges due to the fragile nature of primary MM cells, with significant variability in cell number and from patients. Consequently, we could only use fresh MM patient samples to repeat the key experiments. We obtained three pairs of primary MM patient samples, with one exhibiting negative LILRB1 expression and the others showing positive expression. Our data revealed that primary MM samples with positive LILRB1 expression displayed relatively lower lipid ROS levels compared to MM cells with negative expression (**Fig. R2-1-2 a**). Furthermore, we investigated lipid ROS levels in primary MM samples containing subsets of cells with differing LILRB1 expressions. Our data demonstrated that within the same primary MM patient sample, subsets with lower LILRB1 expression exhibited higher lipid ROS levels compared to MM cells with lower LILRB1 expression (**Fig. R2-1-2 b**), which is consistent with our cell line data showing that LILRB1-KD cells had higher lipid ROS levels than CTR-KD cells. **These data are included in revised Fig. 3 and Supplementary Fig. 3.**

Fig. R2-1-2 Lipid peroxidation of primary MM cells. **(a)** Lipid peroxidation of primary MM cells with negative/positive expression of LILRB1 was measured by BODIPY™ 581/591 C11 dye. **(b)** Lipid peroxidation of primary MM patient samples containing subsets of MM cells with different LILRB1 expression was measured by anti-LILRB1 antibody and BODIPY™ 581/591.

Additionally, we examined the response of primary MM cells to the ferroptosis inducer RSL3. Treating primary MM patient samples with RSL3 for 24 hours revealed varied responses. While LILRB1⁻ patient samples showed induced cell death upon RSL3 treatment (**Fig. R2-1-3 a**), most LILRB1⁺ patient samples exhibited little to no response, with only one sample (#17) showing a significant increase in cell death (**Fig. R2-1-3 b**). Unfortunately, we encountered difficulty in obtaining more LILRB1⁻ patient samples, as most MM patient's samples exhibited positive LILRB1 expression.

In summary, our findings indicated that primary MM cells expressing LILRB1 have lower lipid ROS level and are more resistant to RSL3-induced cell death. **These data are included in revised Fig. 3 and Supplementary Fig. 3.**

Fig. R2-1-3 Cell death induced by RSL3 in primary MM cells with different expression level of LILRB1. **(a,b)** Primary MM cells with negative **(a)** or positive expression **(b)** of LILRB1 were treated with RSL3 and cell death was detected by PI staining with flow cytometry.

2- Figures: the n numbers and number of independent experiments and mice/group should be clearly indicated throughout. As it stands, most figures are n=1 and need to be repeated.

Sorry for the oversight. In our studies, each experiment was repeated at least three times, and each group of animal experiments consisted of more than four mice. We added relevant information to the figure legend.

3- Figure 7: WB images in A and B, do not convincingly indicate that SQLE is higher in LILRB1-KD MM cells, in particular in MOLP-8. The same applies to the WB image shown in Sup Fig. 7B. Overall, I had a hard time interpreting the results shown in Fig 7, which may be due to the lack of my expertise in the pathway shown.

We optimized our experiments and repeated the WB in Figure 7a and b, as well as Supplementary Figure 7a and b. The new data are shown below **(Fig. R2-3-1)**. We believe that these updated data clearly demonstrated an upregulation of SQLE in LILRB1-KD MM cells than CTR-KD MM cells.

The updated WBs are included in revised Fig. 7a-b, and Supplementary Fig. 7a.

Fig. R2-3-1 (a,b) Western blot showing the expression of LDLR, SQLE, GAPDH in CTR-KD or LILRB1-KD MM cells treated with or without cholesterol **(a)** or LDL **(b)**. **(c)** Western blot showing the expression of SQLE and GAPDH in CTR-KD and LILRB1-KD MM cells treated without or with FIN56.

To further elucidate the role of SQLE in ferroptotic cell death, we knocked down SQLE in ARP-1 cells and examined the ferroptotic cell death induced by the ferroptosis inducer RSL3. Our data indicated that SQLE knockdown protected MM cells from RSL3-induced cell death, suggesting that KD of SQLE could protect MM cells from ferroptosis by accumulating squalene **(Fig. R2-3-2)**. We believe that this supplementary data provides additional support to our hypothesis that upregulation of SQLE promotes ferroptosis. **These data are included in revised Supplementary Fig. 7.**

Fig. R2-3-2 Knockdown of SQLE inhibits RSL3-induced ferroptotic cell death in ARP-1 cells. **(a)** Western blot confirming the knockdown efficiency of SQLE. **(b-c)** Cell death induced by RSL3 (400 nm, 10 hours) in CTR-KD or SQLE-KD MM cells was detected by PI staining with flow cytometry. Both the representative histograms **(b)** and summarized data **(c)** are shown. Data are summary of three independent experiments. Statistical significance was

determined by Student t test. ***p< 0.001

4- To confirm the in vitro observations, it would be helpful if MM-bearing mice were treated with RSL3 or similar.

Thanks for the suggestion. We treated MM-bearing mice with RSL3 (15 mg/kg biweekly, the maximum tolerated dose that we had tested in NSG mice). Four groups were established as follows:

- a. CTRL-KD group (n=5): NSG mice injected with CTRL-KD ARP-1 cells, followed by biweekly treatment with the vehicle (corn oil).
- b. CTRL-KD+RSL3 group (n=5): NSG mice injected with CTRL-KD ARP-1 cells, followed by biweekly treatment with RSL3.
- c. LILRB1-KD group (n=5): NSG mice injected with LILRB1-KD ARP-1 cells, followed by biweekly treatment with the vehicle.
- d. LILRB1-KD+RSL3 group (n=5): NSG mice injected with LILRB1-KD ARP-1 cells, followed by biweekly treatment with RSL3.

Both bioluminescent images and summarized data demonstrated that RSL3 could inhibit the progression of LILRB1-KD MM cells in vivo, while having a moderate effect on CTRL-KD MM cells (**Fig. R2-4 a,b**). LILRB1-KD inhibited MM cell progression in vivo, and RSL3 treatment further enhanced the inhibition of MM progression in LILRB1-KD cells, which is consistent with our in vitro data. Tumor burden measured as kappa light chain also supported these findings (**Fig. R2-4 c**). Thus, both our in vitro and in vivo data revealed that knockdown of LILRB1 could sensitize MM cells to ferroptotic cell death and support our conclusion that LILRB1 could protect MM cells from ferroptosis.

These data are included in the revised supplementary Fig. 3.

Fig. R2-4 Effect of RSL3 on CTR-KD and LILRB1-KD MM cells in vivo. NSG mice were injected with 2×10^6 CTR-KD or LILRB1-KD ARP-1 cells, followed by administration of vehicle (veh) or RSL3 (20 mg/kg, ip) biweekly and monitoring of tumor burden ($n=5$). Representative bioluminescent imaging for tumor burden (**a**) and summarized quantification of bioluminescent imaging (**b**) are shown. (**c**) Tumor burden was measured as serum concentration of kappa light chain. Statistical significance was determined by Student t test. * $p < 0.05$; *** $p < 0.001$; n.s., not significant.

Minor comments:

1- Discussion: LILRB1 is an inhibitory receptor with multiple ITIMs, and previous studies indicated that LILRB1 inhibits B cell signaling/functions (reviewed in doi: 10.1172/jci.insight.151553). However, here it is argued that LILRB1 protects the MM cells and provides a survival advantage, so the authors should discuss the discrepancies between these observations.

Thanks for the comments. We agree that it is necessary to discuss the discrepancies between their findings and ours. The review (doi: 10.1172/jci.insight.151553) that you mentioned provides a comprehensive overview of the LILRB1 family, including studies on LILRB1's function in B cells (doi: 10.4049/jimmunol.1300438; doi: 10.1128/CDLI.12.6.705-712.2005.). Different from our observation that LILRB1 promotes the progression of MM cells, previous studies on B cells reported that LILRB1–HLA-G interaction inhibits both naive and memory B cell functions², and LILRB1 downregulates immunoglobulin and cytokine production by human B lymphocytes³. These different observations may be attributed to the potential variation in the

function of LILRB1 in normal B/plasma cells versus MM cells. Additionally, distinct pathways involving LILRB1 may play dominant roles in different types of cells. **We include these discussions in our revised manuscript (line 411-417).**

2- Line 50: provide references.

We provided references (#Human inhibitory leukocyte Ig-like receptors: from immunotolerance to immunotherapy. *JCI Insight*. 2022 Jan 25;7(2):e151553. doi: 10.1172/jci.insight.151553.; #Leukocyte immunoglobulin-like receptor subfamily B: therapeutic targets in cancer. *Antib Ther*. 2021 Feb 9;4(1):16-33. doi: 10.1093/abt/tbab002.) **in our revised version (line 53).**

3- Line 53: include citations showing negative regulation of macrophages by LILRB1 and put more emphasis on immune-evasion.

We included citations demonstrating the negative regulation of macrophages by LILRB1 (#Engagement of MHC class I by the inhibitory receptor LILRB1 suppresses macrophages and is a target of cancer immunotherapy. *Nat Immunol* 19, 76–84 (2018). <https://doi.org/10.1038/s41590-017-0004-z>) and emphasized the immune escape function of LILRB1 **(line 57-61).**

4- Lines 83-84: The grammar for 'longer-survived' and 'poorly survived' should be corrected throughout the text and in the figures.

We replaced “longer survived MM patients” and “poorly survived MM patients” with “MM patients with survival \geq 4years” and “MM patients with survival $<$ 2 years”.

5- Lines 89-90: They should justify why LILRB1 was chosen out of all 20 upregulated genes and link to previous literature and its characteristics, eg, cell surface expression.

Thank you for your suggestions. LILRB1 is a member of the immune inhibitory receptor family LILRBs, which are expressed on the cell surface of immune cells and can contribute to immune evasion⁴; Recently, other members of the LILRB family, such as LILRB3^{5, 6} and LILRB4⁷, have been reported to support the survival of cancer cells; More interestingly, B2M, one of the ligands of LILRB1, was found to be highly upregulated in the serum of most MM patients^{8, 9, 10}, suggesting that LILRB1 may play an important role in MM progression. Consequently, we decided to further investigate the potential of LILRB1 as a target for MM. **We include the above information in our revised version (line 102-108).**

6- Figure 1B: the expression of LILRB1 and other genes seems quite variable among donors, and is up- or down-regulated in poor vs long survivals- the authors should comment on this.

Thank you for the comments. It is common to observe variability in gene expression profiles among different patients, which reflects individual differences. Such heterogeneity is a fundamental aspect of biological diversity, influenced by multiple factors including genetics,

environment, and disease-related factors. The variability in LILRB1 and other gene expressions among patients demonstrated the complexity and diversity of human biology and disease. **We include the following information in our revised manuscript (line 97-99):** 'The expression of the top 20 upregulated genes among patients is shown (Fig. 1b). The variability in gene expression among MM patients demonstrated the complexity and diversity of human biology and disease.'

7- Figure 1: Graphs should be aligned.

Thanks. We have aligned our graphs.

8- Figure 1G and H: How did the define LILRB1 high vs low? The bioinformatics analyses should be described in the M&M section.

Thanks for the comments. In the previous submission, we included the information for defining the expression threshold for LILRB1 in the figure legends. We now added the information on how we defined LILRB1-high and LILRB1-low in the survival analysis to our M&M section in the revised version (**line 522-528**): Overall survival was evaluated in newly diagnosed MM patients based on their LILRB1 expression status. Patients in the top 25% with highest LILRB1 expression were categorized as LILRB1^{high} (red curves), while the remaining patients were classified as LILRB1^{low} (black curves).

9- M&M: Indicate the clone of antibodies used throughout.

We provided the information of antibodies in the report summary.

10- M&M: indicate the sex of mice used.

We added information about mice in M&M section.

11- M&M: provide a summary of the patient characteristics and clinical details, where possible.

Thank you for the comments. We apologize for our inability to obtain patient characteristics and clinical details due to strict privacy regulations in our hospital. As a result, we were unable to conduct a more detailed and in-depth analysis of the patient samples.

12- Figure 3: The size of text in the figure should be increased.

We increased the size of text in our revised figures.

13- Figure 4 and lines 192-193: It is not clear what cell line was used for the co-IP assay in A. Was it LILRB1-transfected 293T cells? If so, this is not the best cell line to use and ideally this should be repeated with MM cell lines.

Thank you for the comment. Specifically, we used the MM cell line ARP-1 for the co-IP assay and sent them for mass-spectrum analysis. We now added this information to both the results section and the M&M section of our revised manuscript.

14- Line 219: replace 'discover' with an alternative word, such as 'investigate'.

Thank you for the suggestion. We now replaced "discover" with "investigate".

15- Sup Fig 8: a descriptive text should be included to summarize the schematic diagram.

Thank you for the suggestion. We now included the following figure legend for Sup Fig 8: In the bone marrow microenvironment of MM patients, ferroptosis-inducing factors such as PUFAs induce lipid peroxidation in MM cells. Normally, LILRB1 expressing on MM cells interacts with LDLRAP1 and LDLR, facilitating LDL/cholesterol uptake and maintaining cell harmony. At this status, the basal squalene level protects MM cells from lipid peroxidation accumulation, preventing ferroptosis and promoting cell survival and MM progression. However, in LILRB1-KD cells, the interaction between LDLR and LDLRAP1 is inhibited, leading to reduced LDL uptake and triggering compensatory cholesterol synthesis by upregulating SQLE expression, which converts squalene to (S)-2,3-epoxysqualene. With decreased squalene protection, MM cells become more susceptible to ferroptosis, inhibiting MM progression.

16- Line 335: correct grammar- 'haven't' to 'have not'.

Thanks for pointing out this mistake. We now replaced "haven't" with "have not".

17- Supp Table 1 and 2: A number of other LILR family members, including LILRB4, appear in the list of genes indicated in S. Table 1. Similarly, LILRB4 comes up in the proteomic analysis. The authors should comment on this, if significant. Do LILRB1 and LILRB2 form heterodimers?

Thank you for the comments. LILRB1 ranked 16th with a p-value of 0.000053, while LILRB4 ranked around 500 with a p-value of 0.0064, and LILRB2 ranked around 6500 with a p-value of 0.047 in Supplementary Table 1. Given that LILRB2 demonstrated a p-value close to 0.05 and a moderate difference in expression, we did not consider it to be a convincing or strongly differentially expressed gene. Additionally, data from the Human Protein Atlas showed low expression of LILRB2 in multiple myeloma cells (<https://www.proteinatlas.org/ENSG00000131042-LILRB2/cell+line#myeloma>). Interestingly, LILRB4 was also found in the pull-down products of the anti-LILRB1 antibody (Supplementary Table 2). Although there may be some sequence similarities between LILRB1 and LILRB4, the manufacturer confirmed that the LILRB1 antibody (ab238145; clone number EPR22861-6; <https://www.abcam.com/products/primary-antibodies/lilrb1-antibody-epr22861-6-ab238145.html>) should not cross-react with LILRB4. Alternatively, it is plausible that LILRB4 may interact with LILRB1. To investigate this possibility, we conducted immunoprecipitation using ARP-1 lysates and an anti-LILRB4 antibody. However, we did not observe any interaction between LILRB4 and LILRB1 (**Fig. R2-17**). We did not employ the anti-LILRB1 antibody for pull-down experiments due to the molecular weight of LILRB4 being approximately 50 kDa. The signal from the heavy

light chain was too strong, resulting in only the heavy light chain band being observed. This suggests that either the interaction between LILRB4 and LILRB1 is too weak to be detected by immunoprecipitation and WB, or there may be no interaction between them. Further investigation is needed to elucidate the connection between LILRB1 and LILRB4.

Fig. R2-17 Interaction between LILRB1 and LILRB4. Immunoprecipitation of LILRB4 was performed by anti-LILRB4 antibody and western blot were conducted to detect LILRB1 and LILRB4 in the pull-down products.

Reviewer #3 (Remarks to the Author):

Major concerns

1. The study, which found that accumulated squalene can inhibit ferroptosis, was limited to lymphomas lacking SQLE and may not be applicable in the context of multiple myeloma. In normal cancer cells, the mevalonate pathway operates rapidly, and as a result, squalene is generally undetectable in the cell. What is the level of SQLE expression in the various MM cells depicted in Fig. 1i? Do the two cells the authors used, ARP-1 and MOLP-8, have high levels of squalene with low SQLE levels of activity? It is necessary to compare the amount of squalene in MM cells and present absolute amounts, particularly in Fig. 7g. In addition, the authors must demonstrate the levels of cholesterol in cells.

Thank you for the insightful questions.

We detected the expression of SQLE in different MM cell lines by WB (**Fig. R3-1 a**). The data showed that MM cell lines expressing LILRB1 exhibited lower SQLE expression, suggesting a potential functional association between LILRB1 and SQLE. Additionally, we assessed squalene levels in various MM cell lines using LC-MS (**Fig. R3-1 b**). While ARP-1 cells, characterized by high LILRB1 expression and low SQLE expression, displayed elevated squalene levels, MOLP-8 MM cells did not exhibit a similar pattern. Intriguingly, our data indicated that squalene levels varied among MM cell lines and did not show a clear correlation with the expression of SQLE or LILRB1. This variability can be attributed to the diverse genetic backgrounds of different MM cell lines, introducing numerous variables influencing squalene regulation.

To further verify our findings, we compared squalene levels in the same MM cell lines with different LILRB1 expressions. The results demonstrated that knocking down LILRB1 led to an increase in SQLE expression and a decrease in squalene levels, aligning with our hypothesis

that LILRB1 knockdown inhibits cholesterol uptake and triggers a compensatory increase in SQLE expression. To enhance clarity, we revised Fig. 7g to present the absolute squalene amounts (**Fig. R3-1 d-e**). Despite potential variations in squalene levels across different samples, it is noteworthy to mention that LILRB1-KD cells consistently exhibited lower squalene levels compared to CTR-KD cells within the same batch or test.

Moreover, we explored cholesterol levels in various MM cell lines, as illustrated in **Fig. R3-1 c**. Interestingly, there were no significant differences in cholesterol levels among different MM cell lines. This observation aligns with the understanding that cholesterol is an essential component of cell membranes, regulated by diverse pathways to maintain a balance. However, upon LILRB1 knockdown, cholesterol uptake was hindered (**Fig. 4 k-p**), prompting a compensatory activation of the cholesterol synthesis pathway (**Fig. 7**) to sustain the elevated demand for cholesterol in proliferating tumor cells. We hope that these clarifications address your queries and provide a more comprehensive understanding of our study.

These data are added to the revised Fig. 7 and Supplementary Fig. 7.

Fig. R3-1 (a) Western blot showing the expression of LILRB1, SQLE, and GAPDH among different MM cell lines. **(b)** Squalene levels among different MM cell lines by HPLC-MS. **(c)** Intracellular cholesterol levels among different MM cell lines by cholesterol assay kit (STA-390, CELL BIOLABS, INC). **(d,e)** CTR-KD or LILRB1-KD MM cells were counted and harvested, followed by the detection of squalene levels in cells by HPLC-MS. For **(c-e)** Data are summary of three biological replicates. Statistical significance was determined by Student t-test. * $p < 0.05$.

2. While the authors used standard ferroptosis inducers like RSL3, FIN56, and erastin, they did not confirm whether ferroptosis-specific inhibitors like ferrostatin-1 or liproxstain-1 can completely rescue cell death and lipid peroxidation. Please also specify how long cells were treated with ferroptosis inducers for cell death and lipid peroxidation analysis.

Thank you for the suggestions. To address this question, we incorporated the ferroptosis-specific inhibitor, liproxstain-1, into our experiments to investigate whether its application could rescue both cell death and lipid peroxidation. Our results demonstrated that liproxstain-1 effectively rescued lipid peroxidation and cell death induced by ferroptosis inducers RSL3 and FIN56 in CTR-KD cells, LILRB1-KD cells, and LILRB1-overexpressing cells, as illustrated in the Fig. R3-2.

Regarding the treatment duration for these experiments, we subjected MM cells to RSL3 for 4 hours to detect lipid peroxidation and 10 hours for cell death detection. For FIN56, MM cells were treated with FIN56 for 18 hours before lipid peroxidation detection, while cell death detection involved a 24-hour treatment. These were determined by our preliminary studies. We incorporated these details into our figure legend for comprehensive clarity.

These data are included in the revised Fig. 3 and Supplementary Fig. 3.

Fig. R3-2 (a-d, i-j) CTR-KD or LILRB1-KD MM cells were treated with ferroptosis inducer RSL3 (400 nM, 4-hours incubation for lipid peroxidation and 10-hours incubation for cell death) or FIN56 (15 μ M, 18-hours incubation for lipid peroxidation and 24-hours incubation for cell death). Lipid peroxidation was measured by BODIPY™ 665/676 dye through flow cytometry, and both the representative histograms (b,d) and the summarized results (a,c) were shown.

Cell death of CTR-KD or LILRB1-KD MM cells was measured by PI staining following flow cytometry (i,j). (e-h, k-l) CTR- or LILRB1-overexpression MM.1R cells were treated with ferroptosis inducer RSL3 (400 nM, 4-hours incubation for lipid peroxidation and 10-hours incubation for cell death) or FIN56 (15 μ M, 18-hours incubation for lipid peroxidation and 24-hours incubation for cell death). Lipid peroxidation was measured by BODIPY™ 665/676 dye through flow cytometry, and both the representative histograms (f,h) and the summarized results (e,g) were shown. Cell death of CTR- or LILRB1-overexpression MM cells was measured by PI staining following flow cytometry (k,l). Data are summary of three biological replicates. Statistical significance was determined by Student t-test. * $p < 0.05$; ** $p < 0.01$; *** $p < 0.001$.

3. Regarding arachidonic acid (AA) treatment for ferroptosis, AA does not normally induce ferroptosis on its own, but it sensitizes cells to ferroptosis when combined with other ferroptosis inducers. Furthermore, when compared to other ferroptosis studies and considering the physiological concentration of AA, 75 nM AA is too low to induce or sensitize cells to ferroptosis. Ferroptosis inhibitors should be used to confirm AA experiments.

Thank you for the valuable comments. We apologize for the oversight in the figure legend. We used 75 μ M AA not 75 nM to treat MM cells. We have thoroughly reviewed the entire manuscript and corrected this error.

In our efforts to validate that AA-induced cell death in MM cells is indeed ferroptotic, we used the ferroptosis inhibitor liproxstatin-1 in treatments with and without AA. Our results showed that liproxstatin-1 not only robustly inhibited AA-induced increase in lipid ROS and cell death in CTR-KD cells, but also effectively inhibited LILRB1 KD-induced increase of lipid ROS and cell death (Fig. R3-3 a-c). Moreover, in LILRB1-overexpressing cells, liproxstatin-1 demonstrated efficacy in inhibiting AA-induced lipid peroxidation and ferroptotic cell death (Fig. R3-3 d-f). These findings provided additional confirmation that AA treatment induces ferroptosis in MM cells, and LILRB1-KD exacerbates ferroptotic cell death.

These data are included in the revised Fig. 3 and Supplementary Fig. 3.

Fig. R3-3 (a-c) CTR-KD or LILRB1-KD MM cells were treated with AA (75 μ M, 4-hours incubation for lipid peroxidation and 10-hours incubation for cell death). Lipid peroxidation was measured by BODIPY™ 665/676 dye through flow cytometry, and both the representative histograms (**b**) and the summarized results (**a**) were shown. Cell death of CTR-KD or LILRB1-KD MM cells was measured by PI staining following flow cytometry (**c**). (**d-f**) CTR- or LILRB1-overexpression MM.1R cells were treated with AA (75 μ M, 4-hours incubation for lipid peroxidation and 10-hours incubation for cell death). Lipid peroxidation was measured by BODIPY™ 665/676 dye through flow cytometry, and both the representative histograms (**e**) and the summarized results (**d**) were shown. Cell death of CTR- or LILRB1-overexpression MM cells was measured by PI staining following flow cytometry (**f**). Data are summary of three biological replicates. Statistical significance was determined by Student t-test. * $p < 0.05$; ** $p < 0.01$; *** $p < 0.001$.

4. If, as the authors claim, LDLR uptake of LDL and cholesterol is the primary mechanism, these experiments should be validated in lipoprotein-deficient serum. Does a lack of lipoproteins cause ferroptosis by upregulating SQLE?

Thanks for your suggestions. We used lipoprotein-deficient serum to validate our hypothesis. When ARP-1 cells were cultured in lipoprotein-free serum, we observed a significant increase in the expression of SQLE (**Fig. R3-4 a**). Intriguingly, the ferroptosis inducer RSL3 induced more cell death in MM cells cultured in normal serum than those cultured in lipoprotein-deficient serum (**Fig. R3-4 b**). Considering that lipoproteins include not only LDL but also HDL, we wondered whether this effect was attributed to the depletion of HDL. To investigate this, we supplemented HDL into the lipoprotein-free serum group and treated the cells with the ferroptosis inducer. The results revealed that the addition of HDL to the lipoprotein-free serum resulted in an increased ferroptotic cell death (**Fig. R3-4 b**). These findings provided additional

support to our proposed mechanism, emphasizing the importance of LDL uptake in mediating the cellular response to ferroptosis.

Taken together, our data demonstrated that LDL protects MM cells from ferroptotic cell death, and the absence of LDL leads to an upregulation of SQLE expression, consequently promoting ferroptotic cell death.

Fig. R3-4 (a) Western blot showing the expression of SQLE in ARP-1 cells cultured in normal medium or lipoprotein-free medium. **(b)** ARP-1 cells were planted in normal medium or lipoprotein-free medium with/without HDL, followed by treatment with ferroptosis inducer RSL3 (400 nM, 10-hours incubation). Cell death was measured by PI staining following flow cytometry. Data are summary of three biological replicates. Statistical significance was determined by Student t-test. **p < 0.01; ***p < 0.001.

5. Indeed, HMGCR is more well-known as the rate-limiting enzyme in the mevalonate pathway than SQLE, but its involvement in this pathway has not been investigated. The authors must provide HMGCR levels upon LILRB1 KD. Furthermore, SQLE upregulation may accumulate CoQ10, which may contribute to ferroptosis suppression. It is difficult to understand how changes in SQLE expression affect each intermediate in the mevalonate pathway without metabolic tracing analysis.

Thanks for the valuable comments.

We performed WB to detect the protein levels of HMGCR and SQLE in CTR-KD and LILRB1-KD cells. The results showed that the expression of HMGCR in CTR-KD and LILRB1-KD MM cells were relatively low and did not show obvious difference, while the expression of SQLE was significantly increased in LILRB1-KD MM cells (**Fig. R3-5-1**).

This data is included in our revised Supplementary Fig. 7.

Fig. R3-5-1 Western blot showing the expression of HMGCR, SQLE, and GAPDH in CTR-KD and LILRB1-KD MM cells.

To confirm the changes of CoQ10 in LILRB1-KD, we detected CoQ10 levels in CTR-KD and LILRB1-KD MM cells by ELISA. Our results showed that the levels of CoQ10 were decreased in LILRB1-KD cells compared to CTR-KD cells (**Fig. R3-5-2 a**). As CoQ10 is also a well-known antioxidant and could help prevent cells from ferroptosis^{11, 12}, the decreased levels of CoQ10 in LILRB1-KD cells could also contribute to their enhanced sensitivity to ferroptotic stress.

We agreed that it would be very helpful if we could do metabolic tracing analysis for all the intermediates in metabolic tracing analysis. However, we met with difficulties in finding a company or core facility that could provide isotope tracing analysis for all the metabolites in the mevalonate pathways. Consequently, we analyzed the most important metabolites in our study: squalene and CoQ10. With ¹³C-glucose as the tracer, we traced the squalene and CoQ10 in ARP-1 CTR-KD cells and LILRB1-KD cells. Our results (**Fig. R3-5-2 b-e**) demonstrated that labeled squalene and CoQ10 were downregulated in LILRB1-KD cells compared to CTR-KD cells, supporting our findings that synthesized squalene and CoQ10 were downregulated by the upregulation of SQLE in LILRB1-KD cells, leading to more ferroptosis.

These data are included in our revised Supplementary Fig. 7.

Fig. R3-5-2 (a) CoQ10 levels of CTR-KD MM cells and LILRB1-KD MM cells were detected by Elisa kit. **(b)** Schematic of analysis of [¹³C]glucose uptake to trace incorporation of ¹³C into newly synthesized squalene and CoQ10. **(c-e)** CTR-KD ARP-1 cells and LILRB1-KD ARP-1 cells were cultured in [¹³C]glucose-containing medium for 24 h and harvested for isotope tracing analysis. **(c)** The labeled fractions of Squalene (M+26) were shown. n=3, biological repeats. **(d)** The labeled fractions of Squalene isotopomers (M+22, ..., M+28) were shown. **(e)** The labeled fractions of CoQ10 (M+16) were. n=2, biological repeats.

6. Because so many drugs can inhibit ferroptosis via non-specific RTA activity, it is difficult to determine whether Ter and NB598's ferroptosis inhibitory effects are due to SQLE inhibition or RTA function. Indeed, the fact that both drugs inhibit ferroptosis in control cells suggests that it is most likely SQLE-independent, or that basal levels of squalene play a role in ferroptosis independent of LILRB1. At the very least, please include the concentrations of the two drugs you used.

Thank you for the comments. We now included the concentrations of the two drugs used in our experiments in the figure legend: specifically, we used 10 μ M terbinafine and 5 μ M NB598.

7. In Fig. 2, the mechanism by which LILRB1 affects tumor growth in vivo is completely unknown in Fig. 2. Is this phenotype caused by ferroptosis or squalene metabolism? The authors have to supply molecular details as well as rescue experiments using ferroptosis inhibitors.

Thank you for the valuable comments and suggestions. First, we analyzed the RNAseq data of CTR-KD and LILRB1-KD MM cells from the bone marrow of in vivo model and identified that LILRB1 deficiency in MM cells activated pathways associated with higher oxidative stress, increased ROS production, and enhanced oxidative stress-induced cell death (**Fig. R3-7-1 a-d**). Second, we examined the levels of lipid ROS in CTR-KD and LILRB1-KD MM cells isolated from murine bone marrow 7- or 15-days post-transplantation of MM cells into NSG mice. The data (**Fig. R3-7-1**) revealed that LILRB1-KD MM cells displayed elevated levels of lipid ROS. **These data are included in our revised Fig. 2 and Supplementary Fig. 2.**

Fig. R3-7-1 (a,b) Pathway analysis of changes in RNAseq data between CTR-KD and LILRB1-KD ARP-1 cells sorted from the bone marrow of tumor burden-NSG mice. **(c)** Heat map of fatty acid metabolism- and ferroptosis-related

genes in RNAseq data of CTR-KD and LILRB1-KD ARP-1 cells sorted from the bone marrow of tumor burden-NSG mice. **(d)** Gene set enrichment analysis of fatty acid metabolism- and ferroptosis-related genes, and heme metabolism-related genes in RNAseq data of CTR-KD and LILRB1-KD ARP-1 cells sorted from the bone marrow of tumor burden-NSG mice. **(e,f)** NSG mice were injected with 5×10^6 CTR-KD or LILRB1-KD ARP-1 cells (n=9). After 7 days (r, n=4) or 15 days (s, n=5) of injection, lipid peroxidation of MM cells in the BM was detected. Statistical significance was determined by Student t-test. *p< 0.05; **p<0.01.

Third, to further validate that increased ferroptosis contributes to the inhibition of MM progression in LILRB1-KD cells in vivo, we treated NSG mice injected with CTR-KD or LILRB1-KD MM cells with the ferroptosis inhibitor liproxstatin-1. The treatment reversed the inhibition of MM progression induced by LILRB1 deficiency in vivo (**Fig. R3-7-2**), confirming that LILRB1 indeed promotes MM progression in vivo by inhibiting MM cell ferroptosis.

These data are included in our revised Fig. 3.

Fig. R3-7-2 NSG mice were injected with 2×10^6 CTR-KD or LILRB1-KD ARP-1 cells through tail vein, followed by administration of vehicle (veh) or ferroptosis inhibitor liproxstatin-1 (lipro, 15 mg/kg, ip) every day and monitor of tumor burden (CTR-KD, n=4; LILRB1-KD, n=4; CTR-KD+liproxstatin-1, n=7; LILRB1-KD, n=7). Representative bioluminescent imaging for tumor burden **(a)** and summarized quantification of bioluminescent imaging **(b)** were shown. **(c)** Tumor burden was measured as serum concentration of κ light chain. Statistical significance was determined by Student t-test. *p< 0.05.

8. The authors must be truthful when citing references. Many references are either irrelevant or have been replaced by review articles. References 10, 39, 40, and 41, in particular, should be replaced with original papers. The references 11 and 12 for AA are not experimental papers. Thanks for pointing out our mistakes. We checked and corrected the references.

Minor concerns

9. In Fig. 3g,m,p,r,t, please display the histogram data for untreated cells. We added the untreated group to the histogram graph (included in the revised Supplementary Fig. 3).

10. In Fig. 3t, Given the difference in area between the two groups, the number of cells measured in each group is likely to be different.

We repeated the experiments and introduced a ferroptosis inhibitor treatment group. Regarding the y-axis scaling settings for histograms, we utilized the modal option, scaling all channels as a percentage of the maximum count. This selection may account for the similarity in the number of cells observed, despite the graphical representation appearing different. For your reference, we attached the raw histograms illustrating the cell numbers below (Fig. R3-10).

Fig. R3-10 Raw histograms of LDL uptake assay in CTR- and LILRB1-overexpressing MM.1R cells.

11. In the legend of Fig. 5, LDL concentration might be “ μM ”. Lipoproteins are complexes comprising lipid complexes (triglycerides, cholesterol, and

phospholipids) and specific proteins, known as apolipoproteins, such as APOB¹³. In human plasma, low-density lipoproteins (LDL) exhibit numerous subspecies with variations in lipid composition¹⁴. Therefore, LDL does not maintain a constant molecular weight, and the quantitative units of human LDL concentration presented in Fig. 5 is expressed in "µg/ml".

12. In Fig. 6, 7, the WB band patterns of SQLE blots are inconsistent. The upper band accumulated specifically upon KD of the gene in some blots, but the opposite result also occurred.

We repeated the WB for SQLE and confirmed that knockdown of LILRB1 resulted in the accumulation of SQLE. Additionally, treatment with LDL/ cholesterol led to downregulation of SQLE, as depicted in the newly presented results in **Fig. R3-12**. SQLE did have two bands in WB as it has been reported that SQLE possesses an endogenous truncated form with a similar abundance to the full-length SQLE¹⁵. This truncated SQLE retains full SQLE activity but is more resistant to cholesterol¹⁵. Our results are consistent with these findings, as we also noted a greater decrease in the full-length SQLE when treated with LDL/cholesterol. This phenomenon may explain occasional discrepancies in the response between the two bands, with the truncated SQLE potentially displaying more moderate changes compared to the full-length SQLE. **These data are included in our revised Fig. 7 and Supplementary Fig. 7.**

Fig. R3-12 (a,b) Western blot showing the expression of LDLR, SQLE, GAPDH in CTR-KD or LILRB1-KD MM cells treated with or without cholesterol **(a)** or LDL **(b)**. **(c)** Western blot showing the expression of SQLE and GAPDH in CTR-KD and LILRB1-KD MM cells treated without or with FIN56.

13. Do not truncate the scale bar on the y-axis, as this magnifies the differences between groups.

We revised our graphs, maintaining the y-axis at its original scale unless adjustments were

deemed necessary.

Reviewer #4 (Remarks to the Author):

1. LILRB1 interacts with LDLR and LDLRAP1 (Fig 4). The authors should map the precise binding sites in these proteins. Does LILRB1 bind to LDLR and LDLRAP1 simultaneously? Are their interactions regulated by LDL?

a. The authors should map the precise binding sites in these proteins.

Thank you for the insightful questions. To address this, we generated various truncated LILRB1 plasmids to explore the regions that may play a crucial role in binding with LDLR or LDLRAP1 (**Fig. R4-1-1 a**).

First, we constructed LILRB1 variants: A4 (1-495aa), A5 (1-540aa), A6 (1-569aa), A7 (1-621aa), and A8 (1-651aa), and then overexpressed them along with LDLR or FLAG-LDLRAP1 to examine their interaction through immunoprecipitation. Our results indicated that LILRB1-A4, A5, A6, A7, and A8 could interact with both LDLRAP1 (**Fig. R4-1-1 b**) and LDLR (**Fig. R4-1-1 c**). This suggests that the binding region may be located within the 1-495aa segment.

Second, to pinpoint the binding sites, we constructed LILRB1-A1 (1-221aa), A2 (1-312aa), A3 (1-460aa), and LILRB1-His-B1 (313-661aa), B2 (462-661aa), B3 (483-661aa), and conducted immunoprecipitation assays with LDLRAP1 or LDLR. Our data revealed that while the interaction between LILRB1-A3, A2, A1, and LDLRAP1 was considerably weaker compared to LILRB1-A4 and full-length LILRB1 (**Fig. R4-1-1 d**), the interaction between LILRB1-B1, B2, and B3 did not significantly differ from that of full-length LILRB1 (**Fig. R4-1-1 e**). This suggests that the cytoplasmic region near the transmembrane domain (483-495aa) is crucial for LILRB1's binding with LDLRAP1.

Regarding LILRB1 binding with LDLR, LILRB1-A1, B2, and B3 showed strong inhibition of interaction compared to other truncated LILRB1 variants, indicating that the extracellular domain, particularly the 221-460aa region of LILRB1, is a potential binding domain with LDLR (**Fig. R4-1-1 f,g**).

These data are included in our revised Supplementary Fig. 4.

Fig. R4-1-1 Interaction of truncated LILRB1 with LDLRAP1 or LDLR **(a)** schematic diagram of the structure of LILRB1 and the designs of different truncated LILRB1 plasmids **(b-g)** immunoprecipitation showing the interaction between LDLR/LDLRAP1 and different truncated LILRB1 in 293T cells overexpressed with LDLR/LDLRAP1 and truncated LILRB1. **(b)** Interaction between FLAG-LDLRAP1 and full length LILRB1 or Truncated LILRB1 A4-A8. **(c)** Interaction between LDLR and full length LILRB1 or Truncated LILRB1 A4-A8. **(d)** Interaction between FLAG-LDLRAP1 and full length LILRB1 or Truncated LILRB1 A1-A4. **(e)** Interaction between FLAG-LDLRAP1 and full length LILRB1 or

Truncated LILRB1 B1-B3. **(f)** Interaction between LDLR and full length LILRB1 or Truncated LILRB1 A1-A4. **(g)** Interaction between LDLR and full length LILRB1 or Truncated LILRB1 B1-B3.

b. Does LILRB1 bind to LDLR and LDLRAP1 simultaneously? Are their interactions regulated by LDL?

Our results demonstrated that LILRB1 is capable of binding to both LDLR and LDLRAP1 (Fig. 4). To investigate whether these interactions are influenced by LDL, we treated cells with LDL for 6 hours and subsequently conducted immunoprecipitation. In our overexpression system of LILRB1 and LDLR in 293T cells, LDL did not alter the interaction between LDLR and LILRB1 (**Fig. R4-1-2 a**). Similarly, in the overexpression system of LILRB1 and LDLRAP1 in 293T cells, LDL did not exhibit a significant influence on the interaction between LILRB1 and LDLRAP1 (**Fig. R4-1-2 b**). Additionally, we examined the interaction in the endogenous system by treating ARP-1 cells with LDL and performing immunoprecipitation. These results also indicated that LDL did not regulate the interaction between LILRB1 and either LDLR or LDLRAP1 (**Fig. R4-1-2 c**).

Fig. R4-1-2 immunoprecipitation showing the influence of LDL on the interaction between LILRB1 with LDLR or LDLRAP1 **(a)** 293T cells were overexpressed with LDLR and LILRB1, followed with/without treatment of LDL. Immunoprecipitation was conducted with anti-LILRB1/anti-IgG antibody and western blot was performed to detect the LDLR and LILRB1 in the pull-down products. **(b)** 293T cells were overexpressed with LDLRAP1 and LILRB1, followed with/without treatment of LDL. Immunoprecipitation was conducted with anti-LILRB1/anti-IgG antibody and western blot was performed to detect the LDLRAP1 and LILRB1 in the pull-down products. **(c)** ARP-1 cells were treated

2. It is important the LILRB1 promotes LDL uptake. Is this process dependent on LDLR and LDLAP1? Since Dab2 is another adaptor protein for LDLR internalization, does Dab2 plays a role in LILRB1-mediated LDL uptake? Can LILRB1 directly mediate LDL uptake? Did the LILRB1 knock-down alter LDLR expression?

Thanks for the valuable questions.

Our data demonstrated that knockdown of LDLR or LDLRAP1 inhibited LDL uptake, indicating the dependence of LDL uptake on LDLR and LDLRAP1 (Fig. 6e,f). We explored whether DAB2 interacts with LILRB1 and contributes to LILRB1-mediated LDL uptake, akin to LDLRAP1. First, we reviewed our mass spectrum data from the pull-down products of the LILRB1 antibody, and

DAB2 was not detected (Supplementary table 2). Second, we performed immunoprecipitation using the lysate of ARP-1 cells but did not find the binding between LILRB1 and DAB2, suggesting that DAB2 does not participate in LILRB1-mediated LDL uptake (**Fig. R4-2 a**).

Apolipoprotein B (APOB) /apolipoprotein E (APOE) are vital components of LDL, facilitating its uptake through interactions with LDLR and participating in LDLR internalization^{16, 17}. To determine whether LILRB1 directly mediates LDL uptake, we conducted immunoprecipitation experiments to explore whether LILRB1 could interact with APOB/APOE, aiming to ascertain whether LDL could directly bind to LILRB1. Our results demonstrated that neither APOB nor APOE showed binding with LILRB1 (**Fig. R4-2 b**). Additionally, our mass spectrum data did not indicate any interaction between apolipoproteins and LILRB1. Given that LILRB4 was reported to interact with APOE⁷, we conducted similar experiments with the paper by flow cytometry to examine whether LILRB1 shares this property. This data further confirmed that LILRB1 does not bind with APOE (Supplementary Fig. 4n). **These data are included in revised Supplementary Fig. 4.**

Taking all these data into consideration, we concluded that the effect of LILRB1 on LDL uptake is dependent on LDLR and LDLRAP1, rather than being mediated directly by LILRB1 itself. LILRB1 promotes LDL uptake by enhancing the interaction between LDLR and LDLRAP1 (Fig. 6a,b).

Fig. R4-2 (a) ARP-1 lysates were used to conduct the immunoprecipitation with anti-LILRB1/anti-IgG antibody and western blot was performed to detect the DAB2 and LILRB1 in the pull-down products. **(b)** ARP-1 lysates were used to conduct the immunoprecipitation with anti-LILRB1/anti-IgG antibody and western blot was performed to detect the APOB and APOE in the pull-down products.

In addition, our data showed that LILRB1-KD did not affect LDLR expression (Fig. 7a,b).

3. The connection between MM poor prognosis, LILRB1 high expression and ferroptosis is weak.

Thank you for the valuable comments. We conducted an additional analysis of patient datasets to compare the expression profiles of ferroptosis-related genes in MM patients with high LILRB1 expression and those with low LILRB1 expression. Our analysis demonstrated that genes associated with the negative regulation of ferroptosis were enriched in MM patients with high LILRB1 expression (**Fig. R4-3**). This data suggests that the expression of LILRB1 may play a

role in protecting MM cells from ferroptosis-induced stress, thereby promoting MM progression and contributing to the poor prognosis of MM patients. **This data is included in revised Fig. 3.** Additionally, our previous analysis of MM patient datasets revealed that patients with recurrent MM exhibited higher expression levels of LILRB1 compared to non-recurrent MM patients. Furthermore, MM patients in advanced stage III displayed significantly elevated LILRB1 expression compared to those in the early stages (I and II). Finally, MM patients with higher LILRB1 expression experienced inferior survival rates than those with lower LILRB1 expression. These findings, **illustrated in Fig. 1**, emphasize that high expression of LILRB1 is closely associated with poor prognosis in MM.

Fig. R4-3 Gene set enrichment analysis of negative regulation of ferroptosis process genes between LILRB1 high expression MM patients and LILRB1 low expression MM patients in Zhan’s MM 2 dataset. MM patients were sorted by the expression level of LILRB1: the top 100 patients with higher expression of LILRB1 were defined as LILRB1 high expression and the bottom 100 patients were defined as LILRB1 low expression.

4. If the hypothesis of this manuscript is correct. The connection between MM and LDLR, PCSK9 and other cholesterol-metabolic genes mutations should be observed. Can the authors provide this information?

We appreciate the valuable comments. We acknowledge the importance of providing additional information about the connection between multiple myeloma (MM) and mutations in cholesterol-metabolic genes to further support our hypothesis. Unfortunately, the MM datasets available to us did not include information on mutations in cholesterol-metabolic genes, making it impossible to perform such an analysis.

Instead of analyzing cholesterol-metabolic gene mutations, we explored the relationship between the expression of LDL/cholesterol uptake-related genes, specifically LDLR and LDLRAP1, and MM patient survival. Our results revealed that MM patients with higher expression of LDLR exhibited lower survival rates compared to those with lower expression of

LDLR (**Fig. R4-4 a**). Similarly, MM patients with higher expression of LDLRAP1 had a poorer prognosis compared to those with lower expression of LDLRAP1 (**Fig. R4-4 b**). Thus, for LDLR and LDLRAP1, which are involved in LDL uptake, higher expression of these genes is related to a poor outcome. These findings align with our LILRB1-related analysis data, providing additional evidence to support our hypothesis. **We include these data in revised Fig. 6 and Supplementary Fig. 6.**

Additionally, we also analyzed the relationship between the expression of PCSK9 and MM patient survival and observed that the lower expression of PCSK9 was related to poorer survival of MM patients (**Fig. R4-4 c**).

Fig. R4-4 (a) Survival of MM patients with high LDLRAP1 (LDLRAP1^{high}) and low LDLRAP1 (LDLRAP1^{low}) expression in Carrasco’s MM dataset, Mulligan’s MM dataset and Zhan’s MM 2 dataset. **(b)** Survival of MM patients with high LDLR (LDLR^{high}) and low LDLR (LDLR^{low}) expression in Mulligan’s MM dataset. **(c)** Survival of MM patients with high PCSK9 (PCSK9^{high}) and low PCSK9 (PCSK9^{low}) expression in Mulligan’s MM dataset. Statistical significance was determined by Log-rank (Mantel-Cox) test and p values are shown.

5. The title of this study should be revised since no clinic applicable targeting approach was used in this study.

We revised the title to “LILRB1 protects human multiple myeloma cells from ferroptosis by maintaining cholesterol homeostasis”.

References

1. Benito-Vicente A, Uribe KB, Jebari S, Galicia-Garcia U, Ostolaza H, Martin C. Familial Hypercholesterolemia: The Most Frequent Cholesterol Metabolism Disorder Caused Disease. *Int J Mol Sci* **19**, (2018).
2. Naji A, *et al.* Binding of HLA-G to ITIM-bearing Ig-like transcript 2 receptor suppresses B cell responses. *J Immunol* **192**, 1536-1546 (2014).
3. Merlo A, *et al.* Inhibitory receptors CD85j, LAIR-1, and CD152 down-regulate immunoglobulin and cytokine production by human B lymphocytes. *Clin Diagn Lab Immunol* **12**, 705-712 (2005).
4. Deng M, *et al.* Leukocyte immunoglobulin-like receptor subfamily B: therapeutic targets in cancer. *Antib Ther* **4**, 16-33 (2021).
5. Wu G, *et al.* LILRB3 supports acute myeloid leukemia development and regulates T-cell antitumor immune responses through the TRAF2-cFLIP-NF-kappaB signaling axis. *Nat Cancer* **2**, 1170-1184 (2021).
6. Huang R, *et al.* LILRB3 Supports Immunosuppressive Activity of Myeloid Cells and Tumor Development. *Cancer Immunol Res* **12**, 350-362 (2024).
7. Deng M, *et al.* LILRB4 signalling in leukaemia cells mediates T cell suppression and tumour infiltration. *Nature* **562**, 605-609 (2018).
8. Bataille R, Magub M, Grenier J, Donnadio D, Sany J. Serum beta-2-microglobulin in multiple myeloma: relation to presenting features and clinical status. *Eur J Cancer Clin Oncol* **18**, 59-66 (1982).
9. Hofbauer D, *et al.* beta(2)-microglobulin triggers NLRP3 inflammasome activation in tumor-associated macrophages to promote multiple myeloma progression. *Immunity* **54**, 1772-1787 e1779 (2021).
10. Bataille R, Grenier J, Sany J. Beta-2-microglobulin in myeloma: optimal use for staging, prognosis, and treatment--a prospective study of 160 patients. *Blood* **63**, 468-476 (1984).
11. Deshwal S, *et al.* Mitochondria regulate intracellular coenzyme Q transport and ferroptotic resistance via STARD7. *Nat Cell Biol* **25**, 246-257 (2023).

12. Hadian K. Ferroptosis Suppressor Protein 1 (FSP1) and Coenzyme Q(10) Cooperatively Suppress Ferroptosis. *Biochemistry* **59**, 637-638 (2020).
13. Mahley RW, Innerarity TL, Rall SC, Jr., Weisgraber KH. Plasma lipoproteins: apolipoprotein structure and function. *J Lipid Res* **25**, 1277-1294 (1984).
14. McNamara JR, Small DM, Li Z, Schaefer EJ. Differences in LDL subspecies involve alterations in lipid composition and conformational changes in apolipoprotein B. *J Lipid Res* **37**, 1924-1935 (1996).
15. Coates HW, Capell-Hattam IM, Brown AJ. The mammalian cholesterol synthesis enzyme squalene monooxygenase is proteasomally truncated to a constitutively active form. *J Biol Chem* **296**, 100731 (2021).
16. Martinez-Olivan J, Arias-Moreno X, Velazquez-Campoy A, Millet O, Sancho J. LDL receptor/lipoprotein recognition: endosomal weakening of ApoB and ApoE binding to the convex face of the LR5 repeat. *FEBS J* **281**, 1534-1546 (2014).
17. Segrest JP, Jones MK, De Loof H, Dashti N. Structure of apolipoprotein B-100 in low density lipoproteins. *J Lipid Res* **42**, 1346-1367 (2001).

REVIEWERS' COMMENTS

Reviewer #1 (Remarks to the Author):

Thank you for the thorough revisions, resulting in many improvements to an already strong body of work.

Reviewer #2 (Remarks to the Author):

The authors have revised the manuscript- it is all satisfactory and I have no further comments. Thank you

Reviewer #3 (Remarks to the Author):

The authors have made extensive revision and have addressed most of the raised issues, which is commendable. However, there are still a few aspects that require attention.

1. While the authors have provided a comprehensive summary of the proposed model in Supplementary Figure 8, they have not included data on the total cholesterol levels maintained upon LILRB1 KD. It is crucial for the authors to present the total cholesterol levels. Additionally, it would be beneficial to demonstrate if de novo cholesterol synthesis is facilitated upon LILRB1 KD using isotope-tracing analysis, as depicted in Supplementary Figure 7.
2. The authors propose that cholesterol itself is not anti-ferroptotic; rather, the expression of SQLE, regulated by cholesterol levels, plays an anti-ferroptotic role through squalene production. Therefore, it would be advantageous to investigate potential changes in squalene levels upon LDL treatment. Including glucose tracing could further elucidate this relationship. Thus, emphasizing the anti-ferroptotic role of squalene in the title or abstract is necessary.
3. Regarding lipoprotein-deficient serum and HDL, as HDL also contains cholesterol, it is challenging to discern the difference between HDL and LDL. However, the observed results align with a recent report indicating that HDL promotes RSL3-induced ferroptosis under lipoprotein deficiency (PMID: 37714840). Therefore, discussing the potential of HDL to play a distinct role in ferroptosis compared to LDL is recommended.
4. Regarding AA levels, 75 μM may not represent a physiological condition and could be considered too high, potentially inducing toxicity. In our conditions, over 50 μM AA induces non-ferroptotic, non-apoptotic, but ROS-dependent cell death in most cancer cells. While MM cells may indeed be more sensitive to ferroptosis, it is necessary for the authors to thoroughly evaluate whether liproxstain-1 fully rescues AA-induced ferroptosis. If not, considering the AA + RSL3 model at 2-10 μM may be more appropriate, but there may be no need to utilize AA as a model in this context. However, if liproxstain-1 does effectively block ferroptosis induced by AA, it is acceptable to emphasize and explain this aspect.

Responses to Reviewers' Comments

We would like to thank the reviewers for careful and thorough reading of this manuscript and for the thoughtful comments and constructive suggestions, which have helped to improve the quality of this manuscript. The corresponding changes and refinements made in the revised paper are summarized in our response below.

Reviewer #3 (Remarks to the Author):

The authors have made extensive revision and have addressed most of the raised issues, which is commendable. However, there are still a few aspects that require attention.

1. While the authors have provided a comprehensive summary of the proposed model in Supplementary Figure 8, they have not included data on the total cholesterol levels maintained upon LILRB1 KD. It is crucial for the authors to present the total cholesterol levels. Additionally, it would be beneficial to demonstrate if de novo cholesterol synthesis is facilitated upon LILRB1 KD using isotope-tracing analysis, as depicted in Supplementary Figure 7.

Thank you for your advice. In Supplementary Fig. 8, we used the same amounts of small circles to illustrate the same level of total cholesterol in both CTR-KD cells and LILRB1-KD cells. However, we understand that this may not be clear enough for readers. To enhance the clarity of our graph, we have made some changes both in the graph and in the figure legend, with the aim of emphasizing that cholesterol levels remained the same between CTR-KD cells and LILRB1-KD cells (shown in Fig.R1). The updated figure and figure legend are now available in Supplementary Fig.8.

Fig.R1 Schematic model of LILRB1's function in MM progression

In the bone marrow (BM) microenvironment of MM patients, ferroptosis-inducing factors such as PUFAs induce lipid peroxidation in MM cells. Normally, LILRB1 expressing on MM cells interacts with LDLRAP1 and LDLR, facilitating LDL/cholesterol uptake and maintaining cell harmony. At this status, the basal squalene level protects MM cells from lipid peroxidation accumulation, preventing ferroptosis and promoting cell survival and MM progression. However, in LILRB1-KD cells, the interaction between LDLR and LDLRAP1 is inhibited, leading to reduced LDL uptake. To maintain cholesterol homeostasis, compensatory cholesterol synthesis is triggered in LILRB1-KD cells by upregulating the expression of cholesterol synthesis rate - limiting enzyme SQLE, which converts squalene, an anti-ferroptotic metabolite, to (S)-2,3-epoxysqualene. With decreased squalene levels, accumulation of lipid peroxidation in LILRB1-KD MM cells makes them more susceptible to ferroptotic cell death, thus inhibiting MM progression.

We agree that it would be beneficial to demonstrate synthesized cholesterol through isotope tracing analysis. Therefore, during our last revision period, we attempted this experiment using [13C]-glucose as the tracer¹. Unfortunately, we were unable to detect labeled cholesterol, although labeled squalene and CoQ10 were detectable. Despite the failure of isotope tracing for de-novo cholesterol synthesis, we believe that another experiment we have done provided

support for the upregulated synthesis of cholesterol in LILRB1-KD cells (Figs. 4m,n): we measured cholesterol uptake by detecting changes in cholesterol levels in the supernatant and found that LILRB1-KD cells exhibited decreased cholesterol uptake. Concurrently, intracellular cholesterol levels were similar in CTR-KD cells and LILRB1-KD cells, indicating that LILRB1-KD cells may synthesize more cholesterol to maintain stable cholesterol levels. We believe that these data can address your concern.

2. The authors propose that cholesterol itself is not anti-ferroptotic; rather, the expression of SQLE, regulated by cholesterol levels, plays an anti-ferroptotic role through squalene production. Therefore, it would be advantageous to investigate potential changes in squalene levels upon LDL treatment. Including glucose tracing could further elucidate this relationship. Thus, emphasizing the anti-ferroptotic role of squalene in the title or abstract is necessary.

Thanks for your comments and suggestions. Regarding squalene levels upon LDL treatment, a published study focused on the cholesterol-dependent degradation of SQLE provides useful information. They labeled cells with [14C]-acetate (which feeds into the beginning of the cholesterol biosynthetic pathway) and found that treatment with LDL or cholesterol led to the accumulation of [14C]-squalene². Additionally, cholesterol treatment was reported to accelerate the degradation of SQLE at the ER membrane by facilitating ubiquitination by the ER-resident MARCH6 E3 ligase and subsequent proteasomal degradation^{2,3}. Moreover, we treated ARP-1 MM cells with LDL or cholesterol and detected squalene levels. Our results showed that LDL/cholesterol treatment could increase the squalene levels (Fig. R2). These data align with our findings that LDL/cholesterol treatment leads to the degradation of SQLE and protects MM cells from ferroptotic cell death. We have included related discussion in our revised manuscript lines 389-391. We also agree that we need to emphasize the anti-ferroptotic role of squalene in the abstract and have updated our abstract in the revised manuscript line 30-31.

Fig.R2 Squalene levels upon LDL/cholesterol treatment. ARP-1 cells were treated with LDL

(a) or cholesterol (b) for 24h and cells were collected and sent for the detection of squalene levels.

3. Regarding lipoprotein-deficient serum and HDL, as HDL also contains cholesterol, it is challenging to discern the difference between HDL and LDL. However, the observed results align with a recent report indicating that HDL promotes RSL3-induced ferroptosis under lipoprotein deficiency (PMID: 37714840). Therefore, discussing the potential of HDL to play a distinct role in ferroptosis compared to LDL is recommended.

Thank you for your suggestions. We agree that it is necessary to discuss the function of HDL on ferroptosis. It is noteworthy that another research group also observed a distinct function of HDL in ferroptosis compared to LDL, aligning with our findings. In their investigation, cells cultured in lipoprotein-deficient human serum exhibited a significant decrease in RSL3-induced ferroptosis⁴. Intriguingly, supplementation with HDL, rather than LDL or VLDL, restored sensitivity to ferroptosis under lipoprotein deficiency, although the underlying mechanisms remain unclear⁴. Both their study and ours indicate that HDL may stimulate ferroptosis, while LDL operates differently in the context of ferroptosis. We appreciate your sharing of this research data and we have included related discussion on lines 394-399.

4. Regarding AA levels, 75 μM may not represent a physiological condition and could be considered too high, potentially inducing toxicity. In our conditions, over 50 μM AA induces non-ferroptotic, non-apoptotic, but ROS-dependent cell death in most cancer cells. While MM cells may indeed be more sensitive to ferroptosis, it is necessary for the authors to thoroughly evaluate whether liproxstain-1 fully rescues AA-induced ferroptosis. If not, considering the AA + RSL3 model at 2-10 μM may be more appropriate, but there may be no need to utilize AA as a model in this context. However, if liproxstain-1 does effectively block ferroptosis induced by AA, it is acceptable to emphasize and explain this aspect.

Thanks for the comments. The levels of AA may vary among different tissues and organs. Since MM resides in the bone marrow, we referred to a study that measured AA concentrations in the bone marrow. Their data indicated that healthy donors typically have an average of around 500 ng/ml (1.6 mM) of AA in human bone marrow, whereas MM patients exhibit lower levels, averaging around 200 ng/ml (656 μM)⁵. Furthermore, AA concentration in the bone marrow of MM patients varies among individuals, ranging from 20 ng/ml (65 μM) to over 500 ng/ml (1.6 mM). Therefore, we believe that 75 μM of AA was a reasonable concentration for MM cells. We utilized AA as a model for ferroptosis to demonstrate that the bone marrow microenvironment experiences ferroptosis-related stress, representing a pathological condition.

Regarding the types of cell death induced by AA on MM cells, we agree with your comment that MM cells may display heightened sensitivity to ferroptosis, as evidenced by our findings that liproxstatin-1 could reverse cell death and lipid ROS induced by AA treatment. Conversely, it is plausible that other cancer cells may exhibit greater resistance to ferroptosis-induced cell stress. Additionally, we appreciate your sharing of the interesting results showing that high-dose AA induces non-ferroptotic, non-apoptotic, but ROS-dependent cell death. We considered that AA-induced cell death could be complex and may vary depending on concentration, treatment duration, and cell type. In our study, we conducted a 4-hour incubation with 75 μ M AA for lipid ROS detection and a 10-hour incubation for cell death detection, which revealed that in MM cells, the early effect induced by AA is ferroptosis. It is also possible that in later stages or in other cell types, alternative forms of cell death may become predominant.

1. Dickson AS, *et al.* A HIF independent oxygen-sensitive pathway for controlling cholesterol synthesis. *Nat Commun* **14**, 4816 (2023).
2. Gill S, Stevenson J, Kristiana I, Brown AJ. Cholesterol-dependent degradation of squalene monooxygenase, a control point in cholesterol synthesis beyond HMG-CoA reductase. *Cell Metab* **13**, 260-273 (2011).
3. Zelcer N, *et al.* The E3 ubiquitin ligase MARCH6 degrades squalene monooxygenase and affects 3-hydroxy-3-methyl-glutaryl coenzyme A reductase and the cholesterol synthesis pathway. *Mol Cell Biol* **34**, 1262-1270 (2014).
4. Oh M, *et al.* The lipoprotein-associated phospholipase A2 inhibitor Darapladib sensitises cancer cells to ferroptosis by remodelling lipid metabolism. *Nat Commun* **14**, 5728 (2023).
5. Panaroni C, *et al.* Multiple myeloma cells induce lipolysis in adipocytes and uptake fatty acids through fatty acid transporter proteins. *Blood* **139**, 876-888 (2022).